# Stress responsive miR-31 is a major modulator of mouse intestinal stem cells during regeneration and tumorigenesis

Yuhua Tian[1†], Xianghui Ma[1†], Cong Lv[1†], Xiaole Sheng[1], Xiang Li[1], Ran Zhao[1], Yongli Song[1], Thomas Andl[2], Maksim V Plikus[3], Jinyue Sun[4], Fazheng Ren[1], Jianwei Shuai[5], Christopher J Lengner[6,7], Wei Cui[8], Zhengquan Yu[1]*

[1]Beijing Advanced Innovation Center for Food Nutrition and Human Health and State Key Laboratories for Agrobiotechnology, College of Biological Sciences, China Agricultural University, Beijing, China; [2]Vanderbilt University Medical Center, Nashville, United States; [3]Department of Developmental and Cell Biology, Sue and Bill Gross Stem Cell Research Center, Center for Complex Biological Systems, University of California, Irvine, Irvine, United States; [4]Institute of Agro-Food Science and Technology, Shandong Academy of Agricultural Sciences, Jinan, China; [5]Department of Physics and State Key Laboratory of Cellular Stress Biology, Innovation Center for Cell Signaling Network, Xiamen University, Xiamen, China; [6]Department of Biomedical Sciences, School of Veterinary Medicine, University of Pennsylvania, Philadelphia, United States; [7]Institute for Regenerative Medicine, University of Pennsylvania, Philadelphia, United States; [8]Institute of Reproductive and Developmental Biology, Faculty of Medicine, Imperial College London, London, United Kingdom

*For correspondence:
zyu@cau.edu.cn

†These authors contributed equally to this work

Competing interests: The authors declare that no competing interests exist.

**Abstract** Intestinal regeneration and tumorigenesis are believed to be driven by intestinal stem cells (ISCs). Elucidating mechanisms underlying ISC activation during regeneration and tumorigenesis can help uncover the underlying principles of intestinal homeostasis and disease including colorectal cancer. Here we show that *miR-31* drives ISC proliferation, and protects ISCs against apoptosis, both during homeostasis and regeneration in response to ionizing radiation injury. Furthermore, *miR-31* has oncogenic properties, promoting intestinal tumorigenesis. Mechanistically, *miR-31* acts to balance input from Wnt, BMP, TGFβ signals to coordinate control of intestinal homeostasis, regeneration and tumorigenesis. We further find that *miR-31* is regulated by the STAT3 signaling pathway in response to radiation injury. These findings identify *miR-31* as a critical modulator of ISC biology, and a potential therapeutic target for a broad range of intestinal regenerative disorders and cancers.
DOI: https://doi.org/10.7554/eLife.29538.001

## Introduction

The intestinal epithelium is one of the most rapidly renewing tissues, undergoing complete turnover in approximately 3 days (*Leblond and Walker, 1956*). This rapid turnover protects against insults from bacterial toxins and metabolites, dietary antigens, mutagens, and exposure to DNA damaging agents including irradiation. Upon insult, the rapid intestinal regeneration is particularly important as impaired regeneration can result in epithelial barrier defects that can lead to rapid dehydration and translocation of intestinal microbiota into the bloodstream. The processes of normal tissue turnover and intestinal regeneration are driven by intestinal stem cells (ISCs) that reside at the bottom of

**eLife digest** Cells lining the inner wall of the gut help to absorb nutrients and to protect the body against harmful microbes and substances. Being on the front line of defense means that these cells often sustain injuries. Specialized cells called intestinal stem cells keep the tissues healthy by replacing the damaged and dying cells.

The intestinal stem cells can produce copies of themselves and generate precursors of the gut cells. They also have specific mechanism to protect themselves from cell death. These processes are regulated by different signals that are generated by the cell themselves or the neighboring cells. If these processes get out of control, cells can easily be depleted or develop into cancer cells. Until now, it remained unclear how intestinal stem cells can differentiate between and respond to multiple and simultaneous signals.

It is known that short RNA molecules called microRNA play an important role in the signaling pathways of damaged cells and during cancer development. In the gut, different microRNAs, including *microRNA-31*,help to keep the gut lining intact. However, previous research has shown that bowel cancer cells also contain high amounts of *microRNA-31*.

To see whether *microRNA-31* plays a role in controlling the signaling systems in intestinal stem cells, Tian, Ma, Lv et al. looked at genetically modified mice that either had too much *microRNA-31* or none. Mice with too much *microRNA-31* produced more intestinal stem cells and were able to better repair any cell damage. Mice without *microRNA-31* gave rise to fewer intestinal stem cellsand had no damage repair, but were able to stop cancer cells in the gut from growing.

The results showed that *microRNA-31* in intestinal stem cells helps the cells to divide and to protect themselves from cell death. It controlled and balanced the different types of cell signaling by either repressing or activating various signals. When Tian et al. damaged the stem cells using radiation, the cells increased their *microRNA-31* levels as a defense mechanism. This helped the cells to survive and to activate repair mechanisms. Furthermore, Tian et al. discovered that *microRNA-31* can enhance the growth of tumors.

These results indicate that *microRNA-31* plays an important role both in repairing gut linings and furthering tumor development. A next step will be to see whether cancer cells use *microRNA-31* to protect themselves from chemo- and radiation therapy. This could help scientists find new ways to render cancerous cells more susceptible to existing cancer therapies.

DOI: https://doi.org/10.7554/eLife.29538.002

crypt and generate the precursors for the specialized differentiated cells (*Barker, 2014*; *Li and Clevers, 2010*).

It has been extensively reported that ISC compartment includes two functionally and molecularly distinct stem cell populations (*Barker, 2014*; *Li and Clevers, 2010*; *Gehart and Clevers, 2015*): The active crypt base columnar (CBC) stem cells (*Sato et al., 2011*), (*Barker et al., 2007*) and a more dormant, reserve ISC population that reside above the crypt base and exhibit no Wnt pathway activity, also referred as +4 cells due to their position at the crypt (*Montgomery et al., 2011*; *Sangiorgi and Capecchi, 2008*; *Tian et al., 2011*; *Takeda et al., 2011*; *Li et al., 2014*; *Yan et al., 2012*). The CBCs often identified and isolated based on the expression of *Lgr5*, a Wnt target gene (*Barker et al., 2007*). During homeostasis, steady-state proliferation of CBCs is driven by extrinsic niche signals – high canonical Wnt activity promotes CBC self-renewal and proliferation (*Barker et al., 2007*; *Miyoshi, 2017*) while BMP signals antagonize it (*Kosinski et al., 2007*). In contrast to the active CBCs, the reserve ISCs represent a slow-cycling population of stem cells that are resistant to high doses of ionizing radiation and appear dispensable for homeostasis (*Sangiorgi and Capecchi, 2008*; *Yousefi et al., 2016*). These reserve ISCs are identified through *CreERT* knockin reporter alleles at the *Bmi1* and *Hopx* loci, as well as by an *Tert-CreERT* transgene (*Montgomery et al., 2011*; *Sangiorgi and Capecchi, 2008*; *Tian et al., 2011*; *Takeda et al., 2011*; *Li et al., 2014*). Reserve ISCs do not have an active Wnt signaling pathway and are refractory to Wnt signals in their resting state (*Takeda et al., 2011*; *Li et al., 2014*; *Li et al., 2016*). Although the activity of the BMP pathway has never been directly examined specifically in reserve ISCs, indirect evidence suggests that it may help to promote their dormancy (*Reynolds et al., 2014*; *He et al., 2004*;

*Kishimoto et al., 2015*). During epithelial regeneration upon stresses, reserve ISCs give rise to Wnt[high] Lgr5[+] CBCs that generate the precursor cells of the specialized differentiated cells (*Tian et al., 2011*; *Takeda et al., 2011*; *Li et al., 2014*). In addition, it has been documented that Lgr5-*CreERT*- or Bmi1-*CreERT*-marked cells can act as the cells of origin of intestinal cancer in mice (*Sangiorgi and Capecchi, 2008*; *Barker et al., 2009*). However, it remains unclear how ISCs differentially sense and respond to multiple signals under both physiological and pathological conditions, and whether these signals contribute to intestinal tumorigenesis.

MicroRNAs represent a broad class of 18–22 nucleotide noncoding RNAs that negatively regulate the stability and translation of target mRNAs. Mounting evidence indicates that microRNAs play important roles in stress-activated pathways (*Leung and Sharp, 2010*; *Mendell and Olson, 2012*; *Emde and Hornstein, 2014*) and in control of somatic stem cell fate and tumorigenesis (*Gangaraju and Lin, 2009*; *Sun and Lai, 2013*; *Yi and Fuchs, 2011*). Hundreds of microRNAs have been identified in the intestinal epithelium (*McKenna et al., 2010*). Global ablation of microRNA activity through genetic deletion of the microRNA processing enzyme *Dicer* demonstrated that microRNAs are critical for homeostasis of intestinal epithelium (*McKenna et al., 2010*). Recently, numerous reports demonstrate that specific microRNAs play important roles in the complex intestinal immune system and in the epithelium during homeostasis including *miR-155*, *miR-29*, *miR-122*, *miR-21*, *miR-146a* and *miR-143/145* (*Runtsch et al., 2014*). Particularly, *miR-143/145* are essential for intestinal epithelial regeneration after injury, acting non cell-autonomously in sub-epithelial myofibroblasts (*Chivukula et al., 2014*), indicating potential importance of microRNA activity in intestinal regeneration.

In the ISC compartment, the function of *miR-31* is of a particular interest, as it becomes overexpressed in colorectal cancer (*Bandrés et al., 2006*; *Cottonham et al., 2010*; *Wang et al., 2009*; *Yang et al., 2013*) and increases during the progression of inflammation-associated intestinal neoplasia (*Olaru et al., 2011*). In addition, it has been reported that *miR-31* is enriched in mammary stem/progenitor cells, suggesting a potential role in somatic stem cells (*Ibarra et al., 2007*). Here we utilized gain- and loss-of-function mouse models to show that a damage-responsive microRNA, *miR-31* drives proliferative expansion of both active and dormant ISCs, and acts as an oncogene promoting intestinal tumorigenesis in different models. Our findings implicated *miR-31* as a potential high-value therapeutic target for a broad range of intestinal regenerative disorders and cancers.

## Results

### *MiR-31* expression pattern in intestine under physiologIcal and pathological conditions

Elevated *miR-31* expression has been previously observed in colorectal cancers (*Bandrés et al., 2006*; *Cottonham et al., 2010*; *Wang et al., 2009*; *Yang et al., 2013*), however its expression in normal intestinal epithelium, particularly in ISCs, remains unclear. To begin addressing a potential role for *miR-31* in the intestinal epithelium and ISCs, first we examined its expression pattern in intestine. *MiR-31* expression levels are the highest in the Lgr5-GFP[high]crypt base columnar stem cells, intermediate in Lgr5-GFP[low] transit-amplifying cell population and the lowest in Lgr5-GFP[neg] populations (*Figure 1A*). Higher level of *miR-31* was also found in Hopx[+] reserve ISCs than that in bulk epithelial cells (*Figure 1A*), based on isolation with *Hopx-CreERT;mTmG* alleles from mice 15 hr after tamoxifen injection. Consistently, in situ hybridization revealed that *miR-31* expression levels are generally higher in the crypts than villi. *MiR-31* is predominantly expressed in the epithelial cells of intestinal crypt, including stem cells and transit amplifying cells (*Figure 1B*). Next, we examined *miR-31* expression in response to intestinal injury. Mice were exposed to 12 Gy γ-IR and then *miR-31* expression was examined at various timepoints during the recovery phase. *MiR-31* levels transiently and markedly drop by 24 hours (coincident with full proliferative arrest/DNA damage response), and then sharply upregulated 48 hours post-γ-IR (during initiation of regenerative proliferation from the radioresistant ISCs), and then return to baseline levels within one week (after full recovery) (*Figure 1C*). In situ hybridization reveals *miR-31* expressing cells to be located in the regenerative foci known to exhibit high Lgr5 expression and Wnt pathway activity (*Figure 1D*). Together, these data suggest a role for this microRNA in ISC-driven regeneration.

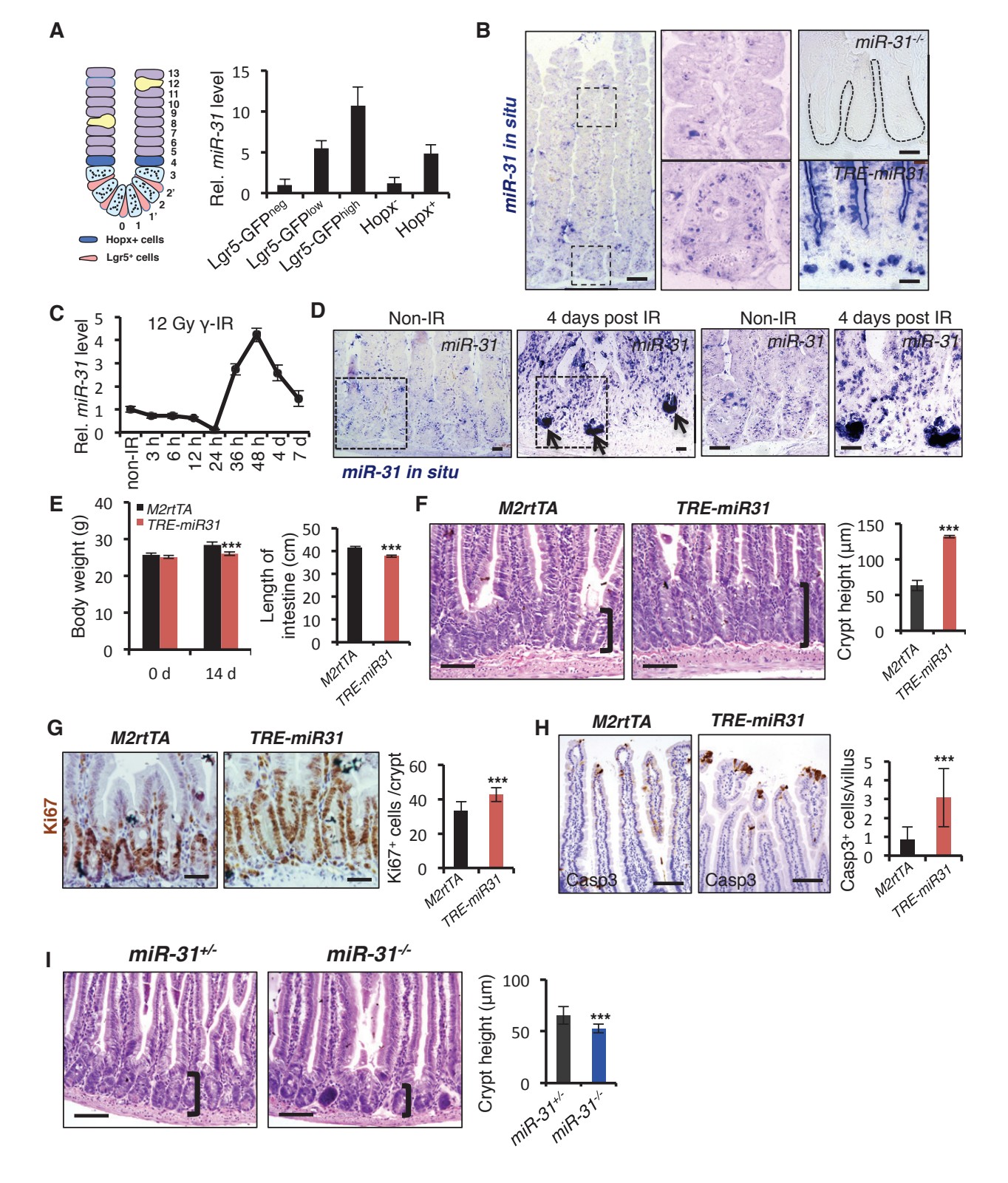

**Figure 1.** *MiR-31* promotes turnover of intestinal epithelial cells. (**A**) Schematic picture of intestinal crypt showing Lgr5[+] CBCs and Hopx[+] cells. qRT-PCR for *miR-31* in Lgr5-GFP[high], Lgr5-GFP[low], Lgr5-GFP[neg], Hopx[-] and Hopx[+] sorted intestinal epithelial cells. n = 4 biological replicates. (**B**) In situ hybridization for *miR-31* in the intestinal epithelium. Left panel, representative low magnification image (Scale bar: 200 µm); Middle panels, high magnification images indicated by dashed boxes in left panel; Right panels (Scale bar: 50 µm), *miR-31* KO intestinal section used as a negative control

*Figure 1 continued on next page*

*Figure 1 continued*

(Top) and *TRE-miR31* (*miR-31* overexpressing) intestinal section used as a positive control (Bottom). (C) qRT-PCR for *miR-31* in the intestinal epithelium after exposure to 12 Gy γ-IR at indicated time points. n = 3 biological replicates. (D) In situ hybridization for *miR-31* in intestines without γ-IR treatment (non-IR), and intestines 4 days after 12 Gy γ-IR. Arrows, *miR-31* positive regenerative foci. Dashes boxes indicate the high magnification images in right panels. Scale bar: 50 μm. (E) Quantification of body weight from *M2rtTA* and *TRE-miR31* mice at the age of 8 weeks before and after Dox treatment for 2 weeks. Quantification of intestine length from *M2rtTA* and *TRE-miR31* mice following 2 week Dox induction. n = 6 biological replicates. ***p<0.001. (F) Representative histologic images showing extension of crypt height in jejunum from *TRE-miR31* mice, and quantification of crypt height from *M2rtTA* and *TRE-miR31* intestine. Both *M2rtTA* and *TRE-miR31* mice were treated with Dox for 2 weeks. n = 3 biological replicates. Scale bar: 50 μm. ***p<0.001. (G) Immunohistochemistry for Ki67 and quantification of Ki67$^+$ cells per crypt in *M2rtTA* and *TRE-miR31* jejunum, showing an expanded proliferative zone in *TRE-miR31* mice following 2 weeks of Dox induction. n = 3 biological replicates. Scale bar: 50 μm. ***p<0.001. (H) Immunohistochemistry for cleaved-Caspase 3 (Casp3) and quantification of Casp3$^+$ cells in the top of intestinal villi from *M2rtTA* and *TRE-miR31* mice following 2 weeks of Dox induction. n = 3 biological replicates. 60 villi were quantified in each mouse. Scale bar: 100 μm. ***p<0.001. (I) Representative histologic images and quantification of crypt height in intestines from *miR-31*$^{+/-}$ and *miR-31*$^{-/-}$ mice at 2 months of age. Brackets mark crypts. Scale bar: 100 μm. n = 3 biological replicates. ***p<0.001.

DOI: https://doi.org/10.7554/eLife.29538.003

The following source data and figure supplements are available for figure 1:

**Source data 1.** Source data for *Figure 1C,E,F,G,H and I*.
DOI: https://doi.org/10.7554/eLife.29538.012
**Source data 2.** Source data for *Figure 1—figure supplements 1–3*.
DOI: https://doi.org/10.7554/eLife.29538.013
**Source data 3.** Source data for *Figure 1—figure supplements 4–7*.
DOI: https://doi.org/10.7554/eLife.29538.014
**Figure supplement 1.** Generation of inducible *TRE-miR-31* transgenic mice, constitutive *miR-31* KO and conditional *miR-31* KO mice.
DOI: https://doi.org/10.7554/eLife.29538.004
**Figure supplement 2.** *MiR-31* induction promotes crypt expansion.
DOI: https://doi.org/10.7554/eLife.29538.005
**Figure supplement 3.** *MiR-31* induction promotes cell proliferation in crypts, and apoptosis at the top of villi.
DOI: https://doi.org/10.7554/eLife.29538.006
**Figure supplement 4.** *MiR-31* induction impairs cell differentiation.
DOI: https://doi.org/10.7554/eLife.29538.007
**Figure supplement 5.** Loss of *miR-31* led to shortened crypt.
DOI: https://doi.org/10.7554/eLife.29538.008
**Figure supplement 6.** Loss of *miR-31* does not affect cell differentiation.
DOI: https://doi.org/10.7554/eLife.29538.009
**Figure supplement 7.** Conditional deletion of *miR-31* resulted in shortened crypt, reduced proliferation and enhanced apoptosis.
DOI: https://doi.org/10.7554/eLife.29538.010
**Figure supplement 8.** *MiR-31* promotes cell turnover from crypt to villi.
DOI: https://doi.org/10.7554/eLife.29538.011

## *MiR-31* promotes intestinal epithelial cell turnover along the Crypt-villus axis

To determine the function of *miR-31* in the mouse intestine, we generated both gain- and loss- of-function mouse models. *MiR-31* gain-of-function was achieved with a targeted, inducible *Rosa26-rtTA;TRE-miR-31* mouse model (*TRE-miR31*) and doxycycline (Dox)-mediated induction of *miR-31* in the intestinal epithelium was validated by qRT-PCR (*Figure 1—figure supplement 1A,B*). For the loss-of-function, we generated constitutive *miR-31* null mice using RNA-guided CRISPR/Cas9 nucleases (*Figure 1—figure supplement 1C*). The 402 bp DNA fragment containing *miR-31* was deleted in the knockout (KO) allele (*Figure 1—figure supplement 1D*), which was validated by sequencing and qRT-PCR (*Figure 1—figure supplement 1E*). We also generated a *Villin-Cre*-mediated intestine-specific conditional *miR-31* null mice (cKO) using traditional homology-directed gene targeting (*Figure 1—figure supplement 1F*). The expression of *miR-31* was markedly reduced in the cKO intestinal epithelium (*Figure 1—figure supplement 1G*). The induction of *miR-31* in *TRE-miR31* intestine and deletion of *miR-31* in KO intestine were also confirmed by in situ hybridization (*Figure 1—figure supplement 1H*).

*MiR-31* induction in response to Dox administration in *TRE-miR31* mice resulted in a significant reduction in body weight after 2 weeks (*Figure 1E*) and intestinal lengths were moderately, but

significantly shorter than controls (*Figure 1E*). Dox treatment of *TRE-miR31* mice for 2 weeks resulted in expansion of intestinal crypts (*Figure 1F*). Unexpectedly villus lengths were mildly shortened, and thus the total length of the crypt-villus was not significantly altered in *TRE-miR31* mice (*Figure 1—figure supplement 2A*). The expanded crypts were also found in the *TRE-miR31* duodenum and ileum (*Figure 1—figure supplement 2B*). The length of intestinal crypts in the control *M2rtTA* mice was not significantly altered at different time points in response to Dox treatment (*Figure 1—figure supplement 2C,D*). In contrast, crypts were significantly expanded in *TRE-miR31* mice after 10 days of Dox treatment, this crypt expansion remained stable for up to 1 year with continuous Dox induction (*Figure 1—figure supplement 2C–E*). Given that crypt elongation reached maximal levels within 2 weeks of Dox induction, we conducted most of the subsequent assays at this time point. More mitotic cells were found in the *TRE-miR31* crypts (*Figure 1G* and *Figure 1—figure supplement 3A,B*), while more apoptotic cells were detected at the top of *TRE-miR31* villi (*Figure 1H* and *Figure 1—figure supplement 3A,B*). The number of Lgr5$^+$ ISCs increased in *TRE-miR31* mice after 10 day Dox treatment, while no significant difference was found between them after 7 days of Dox induction (*Figure 1—figure supplement 3C,D*). In addition, there were fewer differentiated cells including enteroendocrine, goblet and Paneth cells in *TRE-miR31* intestine than the controls (*Figure 1—figure supplement 4A,B*), indicating an impaired cell differentiation. These results suggest that *miR-31* induction accelerates the conveyer-belt movement of proliferative cells exiting the cell cycle and progressing into the villi to ultimately be shed into the lumen, which could comprise the differentiation of specialized intestinal cell types.

Next, we examined the consequence of *miR-31* loss in both *miR-31* germline knockout (KO) and *Villin-Cre*-driven intestinal epithelial conditional KO (cKO) mice. We followed these mice up to six months. Both *miR-31* KO and cKO mice were viable and fertile with no apparent gross phenotypes observed. No differences in the body weight and intestinal length were found between control and *miR-31* KO mice (*Figure 1—figure supplement 5A*), and the transmission of *miR-31* knockout alleles generally followed Mendelian ratios (*Figure 1—figure supplement 5B*). Despite this, loss of *miR-31* led to a significant reduction in crypt height with fewer proliferative cells (*Figure 1I* and *Figure 1—figure supplement 5C,D*). Interestingly, loss of *miR-31* gave rise to a certain number of apoptotic cells throughout the crypt-villus axis, while apoptotic cells are predominantly presented at the tip of control villi and very rare apoptotic cells are presented in crypt-villus axis (*Figure 1—figure supplement 5C,D*). Deletion of *miR-31* also led to increased numbers of enteroendocrine and Paneth cells, while the number of goblet cells remained unaltered in *miR-31* KO intestines (*Figure 1—figure supplement 6A,B*). Moreover, the phenotype of shortened crypts with fewer proliferative cells was also found in cKO intestine (*Figure 1—figure supplement 7A,B*). Loss of *miR-31* gave rise to more apoptotic cells in cKO intestinal epithelium, including in cKO crypts, while cleaved-caspase3$^+$ apoptotic cells were nearly entirely absent from control crypts (*Figure 1—figure supplement 7C,D*). These results suggest that *miR-31* loss functions within intestinal epithelium. We further analyzed DNA synthesis and migration of epithelial cells along the crypt-villus axis after a single pulse of BrdU. Upward movement of BrdU$^+$ cells from crypts to villi was enhanced in *TRE-miR31* mice, and this movement was impaired in *miR-31$^{-/-}$* mice (*Figure 1—figure supplement 8*). Taken together, these data indicate that *miR-31* functions within the intestinal epithelium to maintain a proper balance between stem cell proliferation, differentiation, and epithelial cell death for optimal intestinal homeostasis.

## MiR-31 promotes expansion of Lgr5$^+$ CBC stem cells

Higher expression levels of *miR-31* in Lgr5$^+$ CBCs prompted us to examine its effect on their renewal. Lgr5$^+$ ISC frequency was markedly increased in *TRE-miR31*, and significantly reduced in *miR-31$^{-/-}$* and cKO intestine (*Figure 2A–C* and *Figure 2—figure supplement 1A*). A 1.5 hr pulse of EdU incorporation demonstrated that the frequency of actively proliferating Lgr5-GFP$^+$/EdU$^+$ cells is higher in *TRE-miR31* mice and conversely lower in *miR-31$^{-/-}$* mice (*Figure 2D*). In line with these in vivo findings, *miR-31* induction increased the frequency of budding organoids in vitro, and caused more buds per organoid and more elongated crypts (*Figure 2E* and *Figure 2—figure supplement 1B*). Furthermore, lineage-tracing assay reveals that *miR-31* induction in the intestine increases the height of traced lineages derived from *Lgr5-CreERT*-marked ISCs (*Figure 2F,G* and *Figure 2—figure supplement 1C*). Interestingly, *miR-31* induction significantly repressed *Hopx* expression, while deletion of *miR-31* increased it (*Figure 2H*). Consistently, *miR-31* induction in the intestine repressed lineage tracing from *Hopx-CreERT*-marked reserve ISCs (*Figure 2I*, and *Figure 2—figure*

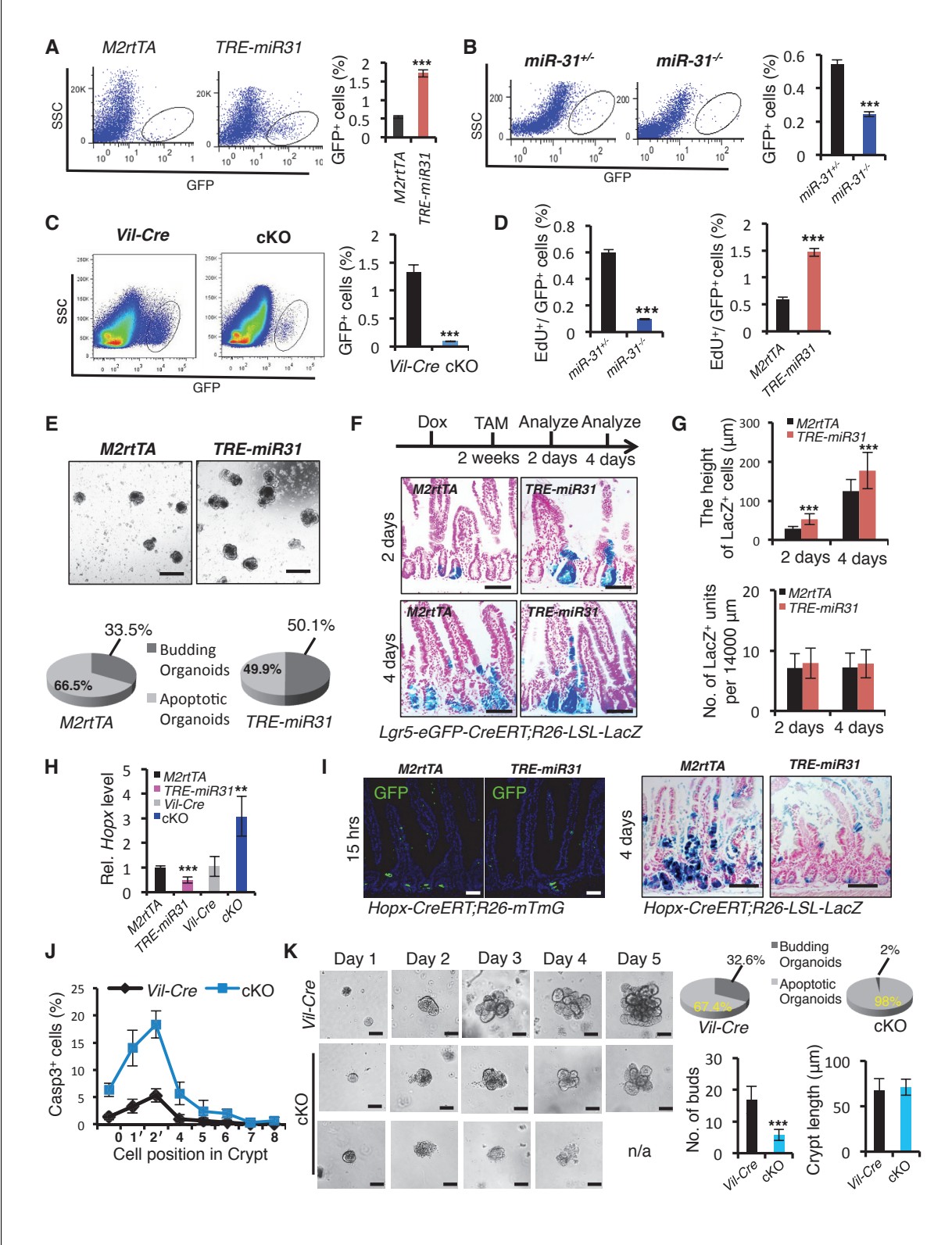

**Figure 2.** *MiR-31* promotes expansion of Lgr5$^+$ CBC stem cells. (**A**) Representative FACS profiles and quantification of GFP positive intestinal epithelial cells (Lgr5-GFP$^+$ cells) from an *Lgr5-eGFP-CreERT* reporter mice crossed with *M2rtTA* (control) and *TRE-miR31* mice. *M2rtTA* (control) and *TRE-miR31* mice were pre-treated with Dox for two weeks. n = 4 biological replicates. ***p<0.001. (**B, C**) FACS profiles and quantification of Lgr5-GFP$^+$ cells from an *Lgr5-eGFP-CreER* reporter mice crossed with *miR-31$^{+/-}$* (control) and *miR-31$^{-/-}$* mice (**B**), or *Vil-Cre* (*Villin-Cre*) and cKO (*Vil-Cre;miR-31$^{fl/fl}$*) mice (**C**).

*Figure 2 continued on next page*

*Figure 2 continued*

n = 4 biological replicates. \*\*\*p<0.001. (D) Assessment of 1.5-hour-pulse EdU incorporation in Lgr5$^+$ CBC cells in *M2rtTA*, and *TRE-miR31* mice following 2 weeks of Dox treatment, and in *miR-31*$^{+/-}$ and *miR-31*$^{-/-}$ intestine. \*\*\*p<0.001. (E) Crypts purified from *M2rtTA* and *TRE-miR31* mice grown in organoid cultures with Dox. Representative gross images of budding organoids, and quantification of budding and apoptotic organoids at day 7. Scale bar: 500 µm. n = 5 technical replicates. (F) X-gal staining showing lineage tracing events from Lgr5$^+$ ISCs. *Lgr5-eGFP-CreERT;R26-LSL-LacZ;TRE-miR31* mice and its control counterpart were pretreated with Dox for 2 weeks, injected with a single dose tamoxifen, and analyzed 2 and 4 days after injection. Scale bar: 100 µm. n = 3 biological replicates. (G) Quantification of the length of LacZ$^+$ cells and LacZ$^+$ units in Panel F. \*\*\*p<0.001. (H) qRT-PCR analysis for *Hopx* in intestines from *M2rtTA*, *TRE-miR31*, *Vil-Cre* and cKO mice. n = 3 biological replicates. \*\*p<0.01; \*\*\*p<0.001. (I) Lineage tracing events from Hopx$^+$ ISCs. *Hopx-CreERT;mTmG;TRE-miR31* mice and their control counterparts were pretreated with Dox for 2 weeks, injected with a single dose of tamoxifen, and analyzed 15 hr after injection. *Hopx-CreERT;R26-LSL-LacZ;TRE-miR31* and their control counterparts were analyzed 4 days after inject with the same treatment. Scale bar: 100 µm. n = 3 biological replicates. (J) Quantification of Cleaved Caspase3$^+$ cells at indicated positions in the intestinal crypts of *Vil-Cre* and *miR-31* cKO mice in *Figure 1—figure supplement 7D*. n = 3 biological replicates, 50 crypts per sample. (K) Crypts purified from *Vil-Cre* and *miR-31* cKO mice grown in organoid cultures at indicated time points. Quantification of budding organoids and apoptotic organoids, budding number and crypt length. n = 3 biological replicates. \*\*\*p<0.001.
DOI: https://doi.org/10.7554/eLife.29538.015

The following source data and figure supplement are available for figure 2:

**Source data 1.** Source data for *Figure 2*.
DOI: https://doi.org/10.7554/eLife.29538.017
**Source data 2.** Source data for *Figure 2—figure supplement 1*.
DOI: https://doi.org/10.7554/eLife.29538.018
**Figure supplement 1.** *MiR-31* promotes ISC expansion.
DOI: https://doi.org/10.7554/eLife.29538.016

supplement 1D,E). In contrast to *miR-31* overexpression, deletion of *miR-31* within intestinal epithelium induced quiescence (residence in G0) in Lgr5-GFP$^+$ cells concomitant to an increase in apoptosis and a decrease in cycling (G1/S/G2/M) (*Figure 2J* and *Figure 2—figure supplement 1F*). In agreement, higher frequency of apoptotic organoids and compromised budding was found in the cKO crypts (*Figure 2K*), and more apoptotic cells were found inside of the cKO organoids (*Figure 2—figure supplement 1G*). Taken together, these data strongly indicate that *miR-31* promotes proliferative expansion of Lgr5$^+$ CBCs, and concomitantly prevents their apoptosis.

## *MiR-31* is critical for intestinal epithelial regeneration following irradiation

The dynamic changes of *miR-31* expression in response to irradiation prompted us to investigate its function during intestinal epithelial injury repair. Intestinal histology of cKO and control *Vil-Cre* mice was comparable two hours after 12 Gy γ-IR (*Figure 3A*). However, by 4 days post-γ-IR, there were significantly fewer regenerative foci and fewer proliferative cells per regenerative focus in cKO mice (*Figure 3A*). Consistently, intestinal regeneration in response to γ-IR was significantly impaired in *miR-31*$^{-/-}$ mice (*Figure 3—figure supplement 1A,B*). Conversely, in the intestine of *TRE-miR31* mice pre-treated for 2 weeks with Dox, there were more regenerative foci with higher numbers of proliferative cells than in the control mice (*Figure 3—figure supplement 1A,B*). These data suggest that *miR-31* is important for intestinal epithelial regeneration in response to irradiation.

To understand the phenotype resulting from *miR-31* modulation, we assayed for apoptotic cells in cKO mice at early stages after irradiation. Loss of *miR-31* increased apoptosis in the crypts 2 and 4 hours post-irradiation prior to any overt histological changes (*Figure 3B*). Quantification of apoptotic cell position analysis reveals that apoptotic events occur with the highest frequently in CBC cells, but are still found in transit-amplifying and +4 zones of cKO crypts, compared to control mice (*Figure 3B*). Further, flow cytometry for live cell and apoptotic markers within the Lgr5-GFP$^+$ population confirmed higher frequency of late apoptotic Lgr5$^+$ cells (AnnexinV$^+$/7AAD$^+$) and lower frequency of early apoptotic Lgr5$^+$ cells (AnnexinV$^+$/7AAD$^-$) and live Lgr5$^+$ cells (AnnexinV$^-$/7AAD$^-$) in cKO mice, relative to controls (*Figure 3—figure supplement 1C*). These data suggest that loss of *miR-31* increases apoptosis of Lgr5$^+$ cells in response to irradiation. Next, we examined its effect on cell proliferation. Cell cycle analysis indicates that more Lgr5-GFP$^+$ cells resided in G0 relative to G1/S/G2/M in cKO mice 2 hours after γ-IR (*Figure 3—figure supplement 1D*). In agreement, expression levels of *Lgr5* were dramatically up-regulated in *TRE-miR31* mice and prominently down-regulated in

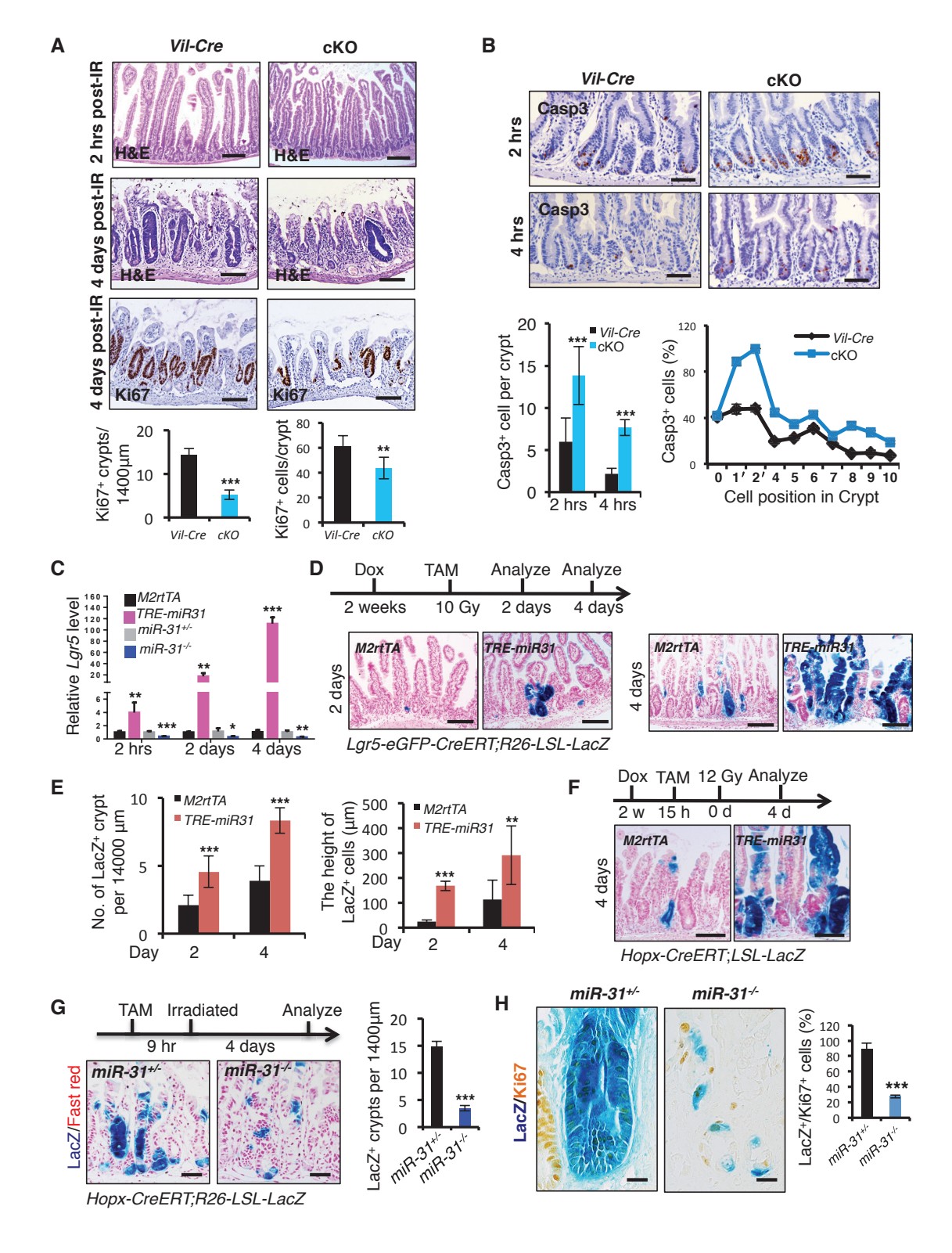

**Figure 3.** Loss of *miR-31* abrogates epithelial regeneration following irradiation. (**A**) Representative images of H&E and/or Ki67 immunohistochemistry from jejunum of irradiated *Vil-Cre* and cKO mice 2 hrs and 4 days post 12 Gy γ-IR. Quantification of Ki67+ regenerative foci per 1400 μm and No. of Ki67+ cells per regenerative focus. Top panel: n = 6 biological replicates; Scale bar: 200 μm. Middle and bottom panels: n = 5 biological replicates; Scale bar: 50 μm. **p<0.01; ***p<0.001. (**B**) Immunohistochemistry for Casp3, quantification of the number of Casp3+ cells in intestinal crypts of *Vil-Cre*

*Figure 3 continued on next page*

*Figure 3 continued*

and cKO mice 2 and 4 hrs post 12 Gy γ-IR. Quantification of Casp3⁺ cells at indicated positions in intestinal crypts of *Vil-Cre* and cKO mice 2 hrs post γ-IR. Scale bar: 50 µm. n = 3 biological replicates, and 50 crypts were quantified in each single mouse. ***p<0.001. (C) qRT-PCR analysis for *Lgr5* in intestines from *M2rtTA*, *TRE-miR31*, *miR-31*$^{+/-}$ and *miR-31*$^{-/-}$ mice 2 hrs, 2 and 4 days post 12 Gy irradiation. *M2rtTA* and *TRE-miR31* mice were pre-treated with Dox for two weeks. n = 3 biological replicates at each time points. *p<0.05; **p<0.01; ***p<0.001. (D) Schematic of *Lgr5-eGFP-CreERT*; *R26-LSL-LacZ* lineage tracing experiment after irradiation. X-gal staining showing lineage tracing events from Lgr5⁺ ISCs. *Lgr5-eGFP-CreERT;R26-LSL-LacZ;TRE-miR31* mice and their control counterparts were pretreated with Dox for 2 weeks, injected with a single dose tamoxifen and then immediately exposed to 10 Gy γ-IR, and analyzed 2 and 4 days after γ-IR. Scale bar: 100 µm. n = 3 biological replicates at each time points. (E) Quantification of LacZ⁺ units and the length of LacZ⁺ cells in Panel D. (F) Schematic of *Hopx-CreERT;R26-LSL-LacZ* lineage tracing experiment. *Hopx-CreERT;R26-LSL-LacZ;TRE-miR31* and their control counterparts were pretreated with Dox for 2 weeks, then injected with a single dose of tamoxifen, and then irradiated 15 hrs after injection and analyzed 4 days after irradiation. Representative images of LacZ staining in *M2rtTA* and *TRE-miR31* intestine 4 days post 12 Gy γ-IR. Scale bar: 50 µm. Statistics of LacZ⁺ regenerative foci were shown in *Figure 3—figure supplement 1E*. n = 3 biological replicates. (G) Schematic of *Hopx-CreERT;R26-LSL-LacZ* lineage tracing experiment. Representative images of LacZ staining in *miR-31*$^{+/-}$ and *miR-31*$^{-/-}$ intestine 4 days post 12 Gy γ-IR. Scale bar: 50 µm. Statistics of LacZ⁺ regenerative foci. n = 3 biological replicates. (H) Representative images of LacZ (blue) and Ki67 (yellow) immunostaining in *miR-31*$^{+/-}$ and *miR-31*$^{-/-}$ intestinal crypts, and statistics of percentage of LacZ⁺/Ki67⁺cells in regenerative foci. Scale bar: 25 µm. n = 3 biological replicates. ***p<0.001.

DOI: https://doi.org/10.7554/eLife.29538.019

The following source data and figure supplement are available for figure 3:

**Source data 1.** Source data for *Figure 3*.
DOI: https://doi.org/10.7554/eLife.29538.021
**Source data 2.** Source data for *Figure 3—figure supplement 1*.
DOI: https://doi.org/10.7554/eLife.29538.022
**Figure supplement 1.** *MiR-31* is required for intestinal epithelial regeneration in response to γ-IR.
DOI: https://doi.org/10.7554/eLife.29538.020

*miR-31*$^{-/-}$ mice at multiple time points after irradiation (*Figure 3C*), and consequently *miR-31* induction promoted lineage regeneration from Lgr5⁺ cells in response to irradiation (*Figure 3D,E*).

Reserve ISCs, marked either by *Bmi1-CreER* or *Hopx-CreER* reporters, have been reported to resist high dose of radiation, being able to replenish the depleted CBC compartment and regenerate the epithelium after irradiation (*Sangiorgi and Capecchi, 2008*; *Tian et al., 2011*; *Takeda et al., 2011*; *Yan et al., 2012*), (*Yousefi et al., 2016*). Thus, we examined the response of *Hopx-CreER*-marked reserve ISCs to 12 Gy γ-IR upon *miR-31* induction and deletion. Lineage-tracing assay revealed that *miR-31* induction promoted epithelial regeneration from the Hopx⁺ reserve stem cells (*Figure 3F* and *Figure 3—figure supplement 1E*). Conversely, the number and the size of regenerative foci originating from *Hopx-CreER;Rosa26-LoxP-Stop-LoxP-LacZ*-marked cells were markedly reduced in *miR-31*$^{-/-}$ mice (*Figure 3G*). In line with this, the frequency of LacZ⁺/Ki67⁺ cells was significantly lower in *miR-31*$^{-/-}$ mutants compared to controls (*Figure 3H*). Taken together, *miR-31* deficiency-mediated the reduction in proliferation and increase in apoptosis within both CBC and reserve ISC compartments can account for the impaired regeneration of *miR-31* null intestine.

## *MiR-31* activates the Wnt pathway and represses the BMP and TGFβ pathways

Canonical Wnt pathway activity is a major driving force for self-renewal of CBCs and epithelial regeneration after injury (*Clevers et al., 2014*), and, thus we examined the effect of *miR-31* on Wnt activity. We utilized *Axin2-LacZ* Wnt reporter mice, which act as a broad readout for canonical Wnt activity, and normally showed its activity to be restricted to the base of crypts in control mice, as expected (*Figure 4A*) (*Davies et al., 2008*). In contrast, Wnt pathway activity was strikingly absent from CBCs of *miR-31*$^{-/-}$ crypts, appearing only faintly above the crypt base in the early TA zones (*Figure 4A*). Conversely, Wnt activity was expanded in *TRE-miR31* crypts (*Figure 4A,B*). In agreement, the number of nuclear β-Catenin-positive cells was significantly reduced in *miR-31*$^{-/-}$ intestinal crypts at 2 and 4 months of age (*Figure 4—figure supplement 1A*). Conversely, they increase in *TRE-miR31* crypts 14 days and 2 months after Dox induction (*Figure 4—figure supplement 1B*). Consistently, the expression levels of *Ctnnb1* (encoding β-Catenin) and the Wnt targets, *Ccnd1* (encoding Cyclin D1), *Myc* and *Axin2* were significantly reduced in *miR-31*$^{-/-}$ intestine both at the RNA and protein levels (*Figure 4C,D*). In contrast, expression levels of the above genes were enhanced in *TRE-miR31* intestinal epithelium following 2 weeks of Dox induction (*Figure 4E,F*). The

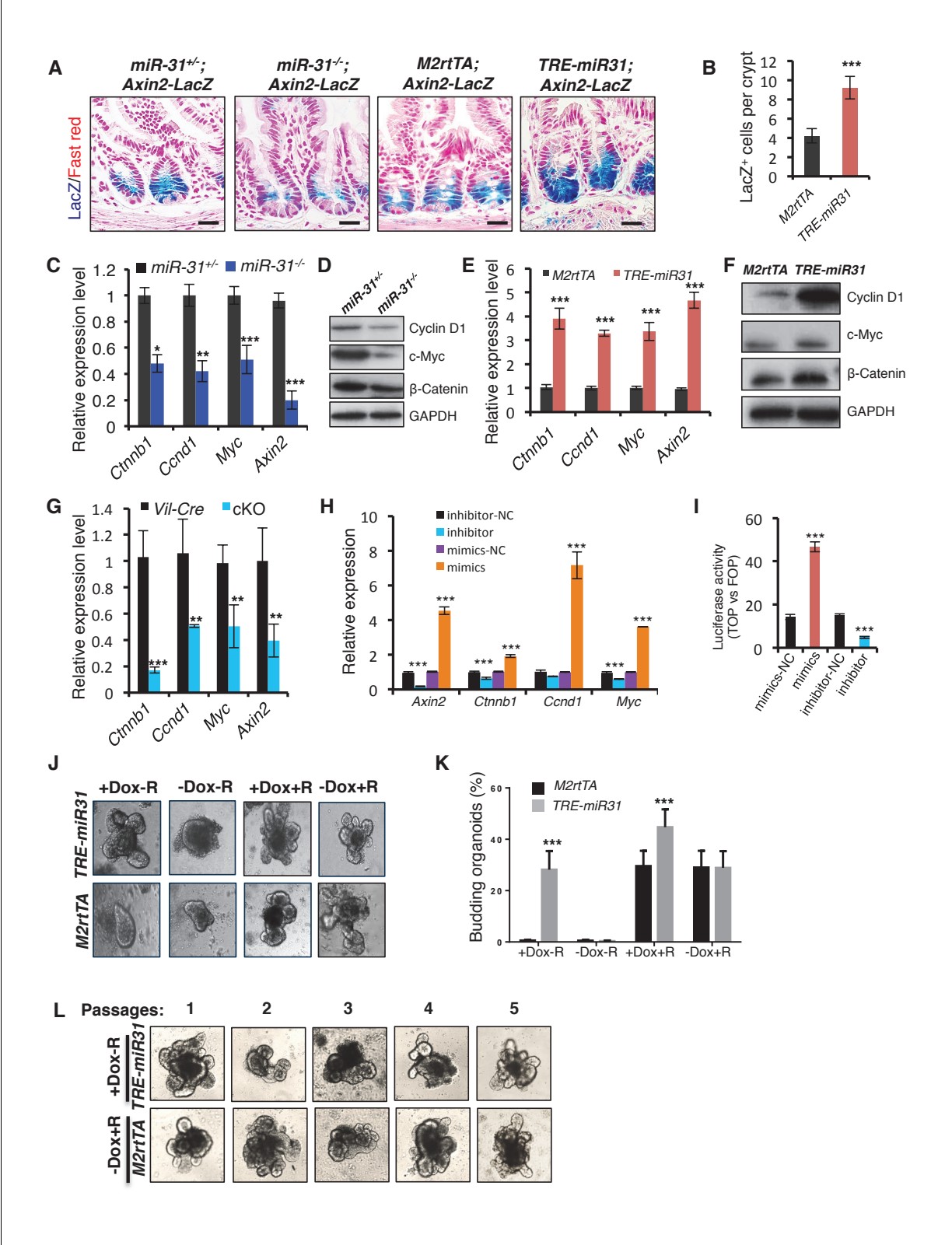

**Figure 4.** *MiR-31* activates Wnt pathway activity. (**A**) Wnt activity was evaluated by *Axin2-LacZ* reporter activity in *M2rtTA* and *TRE-miR31* intestine following 2 week Dox induction, and in *miR-31*$^{+/−}$ and *miR-31*$^{−/−}$ intestine. Blue, LacZ signals. n = 3 biological replicates. Scale bar: 25 μm. (**B**) Quantification of LacZ$^+$ cells per crypt in *M2rtTA* and *TRE-miR31* mice. ***p<0.001. (**C**) qRT-PCR analysis for *Ctnnb1* (encoding β-Catenin), *Ccnd1* (encoding Cyclin D1), *Myc*, and *Axin2* in *miR-31*$^{+/−}$ and *miR-31*$^{−/−}$ intestine. *p<0.05; **p<0.01; ***p<0.001. (**D**) Western blotting for Cyclin D1, c-Myc
*Figure 4 continued on next page*

*Figure 4 continued*

and β-Catenin in *miR-31*$^{+/-}$ and *miR-31*$^{-/-}$ intestine. GAPDH was used as a loading control. (**E**) qRT-PCR for *Ccnd1*, *Myc*, *Axin2* and *Ctnnb1* in intestine from *M2rtTA* and *TRE-miR31* mice following 2 weeks of Dox induction. ***p<0.001. (**F**) Western blotting for Cyclin D1, c-Myc, and β-Catenin in intestine from *M2rtTA* and *TRE-miR31* mice following 2 weeks of Dox induction. (**G**) qRT-PCR for *Ctnnb1*, *Ccnd1*, *Myc*, and *Axin2* in intestine from *Vil-Cre* and cKO mice. n = 4 biological replicates. **p<0.01; ***p<0.001. (**H**) qRT-PCR for *Axin2*, *Ccnd1*, *Myc*, and *Ctnnb1* in HCT116 colon cancer cells treated with *miR-31* inhibitor and negative control (NC, Scramble RNA), as well as *miR-31* mimics and negative control (NC, Scramble RNA) for 24 hrs. ***p<0.001. (**I**) Luciferase activity of TOPflash versus FOPflash in HCT116 cells treated with *miR-31* inhibitor and negative control (NC, Scramble RNA), as well as *miR-31* mimics and negative control (NC, Scramble RNA) for 24 hrs. n = 3 technical replicates. ***p<0.001. (**J**) Representative images of organoids cultures from purified *M2rtTA* and *TRE-miR31* crypts at indicated conditions. R; R-Spondin. n = 3 biological replicates. (**K**) Quantification of budding organoids in Panel J. ***p<0.001. (**L**) Representative images of organoids cultures from purified *M2rtTA* and *TRE-miR31* crypts at serial passages. *M2rtTA* organoids were cultured with R-Spondin; *TRE-miR31* organoids were cultured with Dox and without R-Spondin. n = 4 biological replicates.
DOI: https://doi.org/10.7554/eLife.29538.023

The following source data and figure supplement are available for figure 4:

**Source data 1.** Source data for *Figure 4*.
DOI: https://doi.org/10.7554/eLife.29538.025
**Source data 2.** Source data for *Figure 4—figure supplement 1*.
DOI: https://doi.org/10.7554/eLife.29538.026
**Figure supplement 1.** *MiR-31* activates Wnt signaling pathway.
DOI: https://doi.org/10.7554/eLife.29538.024

reduction in *Ctnnb1* and Wnt targets was further confirmed in conditional *miR-31* KO intestine (*Figure 4G*). To test whether Wnt activity is directly impacted by *miR-31*, we analyzed the effects of gain- and loss-of-function of *miR-31* on expression of Wnt target genes in HCT116 human colorectal carcinoma cells. *Ccnd1*, *Ctnnb1*, *Myc* and *Axin2* were markedly increased in *miR-31* over-expressing cells, relative to controls (*Figure 4H*). Conversely, these genes were downregulated upon *miR-31* inhibition (*Figure 4H*). Considering that HCT116 cells are heterozygous for a β-Catenin gain-of-function mutation at the Gsk3b target site S45 (Ctnnb1$^{+/S45mt}$) (*Ilyas et al., 1997*), (*Kaler et al., 2012*), we examined β-Catenin protein levels. Consistently, β-Catenin was up-regulated in the presence of *miR-31* mimics, and down-regulated upon *miR-31* inhibition (*Figure 4—figure supplement 1C*). The Wnt reporter (Topflash/Fopflash) assay using HCT116 cells further confirmed that *miR-31* induction enhanced Wnt activity, while inhibition of *miR-31* repressed it (*Figure 4I*). To test the functional relevance of *miR-31* potentiation of canonical Wnt activity, we cultured organoids with varying combinations of *miR-31* induction and R-spondin, the Lgr5 ligand. Wnt activation by R-spondin is critical for normal organoid growth and budding (*Sato et al., 2011*). Interestingly, we observed that *miR-31* induction via *TRE-miR31* was sufficient to maintain crypt organoid growth and budding in the absence of R-spondin (*Figure 4J,K*) and that the Dox-treated *TRE-miR31* organoids can be normally passaged at least five times, similar to the organoids cultured with R-spondin (*Figure 4L*). Together, these findings demonstrate that *miR-31* activates the canonical Wnt signaling in the crypts of small intestine.

BMP and TGFβ pathways are known to inhibit the canonical Wnt pathway, inhibiting proliferation and promoting intestinal progenitor differentiation (*Reynolds et al., 2014*; *He et al., 2004*; *Furukawa et al., 2011*). We thus examined the effects of *miR-31* on BMP and TGFβ signals. BMP-specific Smad1/5/8 and TGFβ-specific Smad2/3 phosphorylation were significantly increased in *miR-31*$^{-/-}$ intestine (*Figure 5A* and *Figure 5—figure supplement 1A*), and downregulated in *TRE-miR31* intestine (*Figure 5A* and *Figure 5—figure supplement 1B*), suggesting an inhibitory effect of *miR-31* on BMP and TGFβ signaling pathways. Consistently, we observed a significant increase on the expression of BMP target genes including *Id1*, *Id2*, *Id3*, *Msx1*, *Msx2* and *Junb* and TGFβ target genes *Cdkn1c* (*p57*), *Cdkn1a* (*p21*), *Cdkn2a* (*p16*), and *Cdkn2b* (*p15*) in *miR-31*$^{-/-}$ intestine (*Figure 5B*). Conversely, BMP and TGFβ targets were repressed upon forced expression of *miR-31* in *TRE-miR31* intestine following 2 weeks of Dox induction (*Figure 5C*). The upregulation of BMP and TGFβ targets was further confirmed upon conditional *miR-31* deletion in cKO intestine (*Figure 5D,E*). BMP-specific Smad1/5/8 and TGFβ-specific Smad2/3 phosphorylation were also increased in *miR-31* cKO cultured organoids (*Figure 5—figure supplement 1C*). Further, we examined the effect of *miR-31* on BMP and TGFβ signaling in HCT116 colorectal cancer cells. These cells carry biallelic mutations in the *Tgfbr2* gene, but still express functional TGFBR2 proteins and

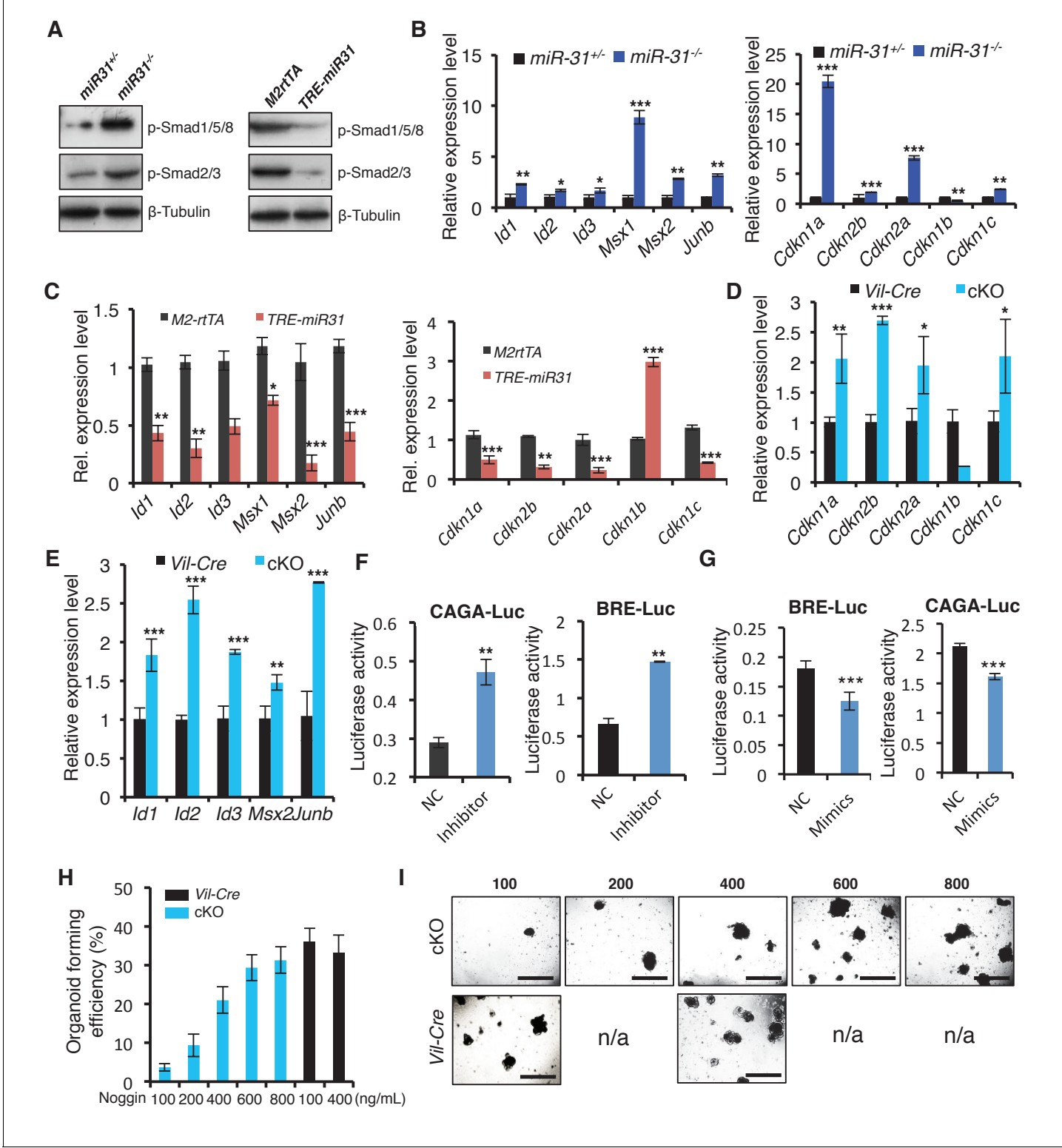

**Figure 5.** *MiR-31* represses BMP/TGFβ signaling pathways. (**A**) Western blotting for p-Smad1/5/8 and p-Smad2/3 in *miR-31*$^{+/-}$, *miR-31*$^{-/-}$, *M2rtTA* and *TRE-miR31* intestine. Both *M2rtTA* and *TRE-miR31* mice were treated with DOX for 2 weeks. β-Tubulin was used as a loading control. (**B**) qRT-PCR analysis for BMP downstream genes, *Id1*, *Id2*, *Id3*, *Msx-1*, *Msx-2* and *Junb*, and TGFβ downstream genes, *Cdkn1c* (p57), *Cdkn1a* (p21), *Cdkn2a* (p16), *Cdkn2b* (p15) and *Cdkn1b* (p27) in *miR-31*$^{+/-}$ and *miR-31*$^{-/-}$ intestine. *p<0.05; **p<0.01; ***p<0.001. (**C**) qRT-PCR analysis for BMP downstream genes, *Id1*, *Id2*, *Id3*, *Msx-1*, *Msx-2* and *Junb*, and TGFβ downstream genes, *Cdkn1c*, *Cdkn1a*, *Cdkn2a*, *Cdkn2b* and *Cdkn1b* in *M2rtTA* and *TRE-miR31* intestine
*Figure 5 continued on next page*

Figure 5 continued

following 2 weeks of Dox induction. **p<0.01; ***p<0.001. (D) qRT-PCR analysis for TGFβ downstream genes, *Cdkn1c*, *Cdkn1a*, *Cdkn2a*, *Cdkn2b* and *Cdkn1b* in intestine from *Vil-Cre* and cKO mice. *p<0.05; **p<0.01; ***p<0.001. (E) qRT-PCR analysis for BMP downstream genes, *Id1*, *Id2*, *Id3*, *Msx2* and *Junb* in *Vil-Cre* and cKO intestine. **p<0.01; ***p<0.001. (F and G) HEK293T cells were transfected with CAGA- or BRE- luciferase reporter vector, combined with scramble RNA (negative control, NC) or anti-miR-31 (miR-31 inhibitors) (F), or scramble RNA (negative control, NC) and *miR-31* mimics (G) for 24 hrs and then harvested for luciferase activity determination. n = 3 technical replicates. **p<0.01; ***p<0.001. (H) Quantification of organoid forming efficiency (budding organoids per 100 crypts) after *Vil-Cre* or cKO crypts cultured with noggin at indicated concentrations for 4 days. n = 3 technical replicates. (I) Representative images of organoids from *Vil-Cre* and cKO crypts cultured with noggin at indicated concentrations (100, 200, 400, 600 and 800 ng/mL) for 4 Days in Panel H.

DOI: https://doi.org/10.7554/eLife.29538.027

The following source data and figure supplement are available for figure 5:

**Source data 1.** Source data for *Figure 5*.
DOI: https://doi.org/10.7554/eLife.29538.029
**Figure supplement 1.** *MiR-31* represses BMP and TGFβ signaling pathways.
DOI: https://doi.org/10.7554/eLife.29538.028

respond to TGFβ (*de Miranda et al., 2015*). In line with the in vivo findings, we found down-regulation of p-Smad2/3 and p-Smad1/5/8 in HCT116 cells treated with *miR-31* mimics, and their up-regulation in cells treated with *miR-31* inhibitor (*Figure 5—figure supplement 1D*). Luciferase assays using BMP- and TGFβ-responsive luciferase reporters, *BRE-Luc* and *CAGA-Luc*, respectively, revealed that inhibition of *miR-31* resulted in significant increases in luciferase activities, and that *miR-31* mimics decreased them (*Figure 5F,G*). More importantly, increasing concentrations of the BMP inhibitor Noggin in organoid culture was able to rescue the budding defect in *miR-31* cKO organoids in a dose-dependent manner (*Figure 5H,I*). Together, these data suggest that *miR-31* promotes ISC proliferation possibly through repressing BMP and TGFβ signaling pathways in a cell-autonomous manner.

## Identification of direct targets of *miR-31*

To understand how *miR-31* regulates Wnt, BMP and TGFβ pathways, we analyzed *miR-31* binding sites in 3'UTRs of transcripts encoding for regulators of these pathways. Genes containing *miR-31* binding sites include Wnt antagonists *Axin1*, *Gsk3b*, and *Dkk1*, along with transcripts containing BMP/TGFβ signaling pathway components such as *Smad3*, *Smad4*, *Bmpr1a* and *Tgfbr2* (*Figure 6—figure supplement 1A*). The expression of *Axin1*, *Gsk3b*, *Dkk1*, *Smad3*, *Smad4*, *Bmpr1a* and *Tgfbr2* was significantly upregulated in *miR-31*$^{-/-}$ intestine (*Figure 6A*) and remarkably downregulated in *TRE-miR31* intestine following Dox induction (*Figure 6B*), suggesting that they are negatively regulated by *miR-31*. The upregulation of these putative target genes was further confirmed in conditional *miR-31* KO intestine (*Figure 6C*). *Axin1*, *Gsk3b*, *Dkk1*, *Bmpr1a* and *Smad4* were selected for further validation at protein level (*Figure 6D,E* and *Figure 6—figure supplement 2A–C*) and in organoids cultured from *miR-31* cKO mice (*Figure 6—figure supplement 3A*). This effect was further confirmed in HCT116 cells with *miR-31* modulation (*Figure 6—figure supplement 3B*). Next, we validated the direct repression of target transcripts by *miR-31* activity using WT-3'UTR-luciferase constructs for *Axin1*, *Gsk3b*, *Dkk1*, *Bmpr1a*, *Smad3* and *Smad4*. Mutation of the *miR-31* 3'UTR binding site in these constructs abrogated this repression (*Figure 6F* and *Figure 6—figure supplement 1B*). Furthermore, RNA crosslinking, immunoprecipitation, and RT-PCR (CLIP-PCR) assays with Ago2 antibodies confirmed that transcripts of *Axin1*, *Dkk1*, *Gsk3b*, *Smad3*, *Smad4* and *Bmpr1a* were highly enriched in Ago2 immunoprecipitates, and that increasing *miR-31* activity augmented their enrichment (*Figure 6G*), providing evidence that *miR-31* directly binds to these transcripts. Taken together, these findings indicate that *Axin1*, *Gsk3b*, *Dkk1*, *Smad3*, *Smad4*, and *Bmpr1a* transcripts are the direct targets of *miR-31*. Next, we asked whether these targets functionally contribute to impaired regeneration in *miR-31*$^{-/-}$ mice. Derepression of these target transcripts was observed in *miR-31*$^{-/-}$ intestine after irradiation (*Figure 6H,I*). As a consequence, Wnt activity was reduced, while the BMP and TGFβ activities were increased in *miR-31*$^{-/-}$ intestine, evidenced by β-Catenin, p-Smad1/5/8 and p-Smad2/3 immunohistochemistry assays (*Figure 6J*). Considering that intestinal regeneration following irradiation requires Wnt hyperactivity (*Davies et al., 2008*), and that BMP

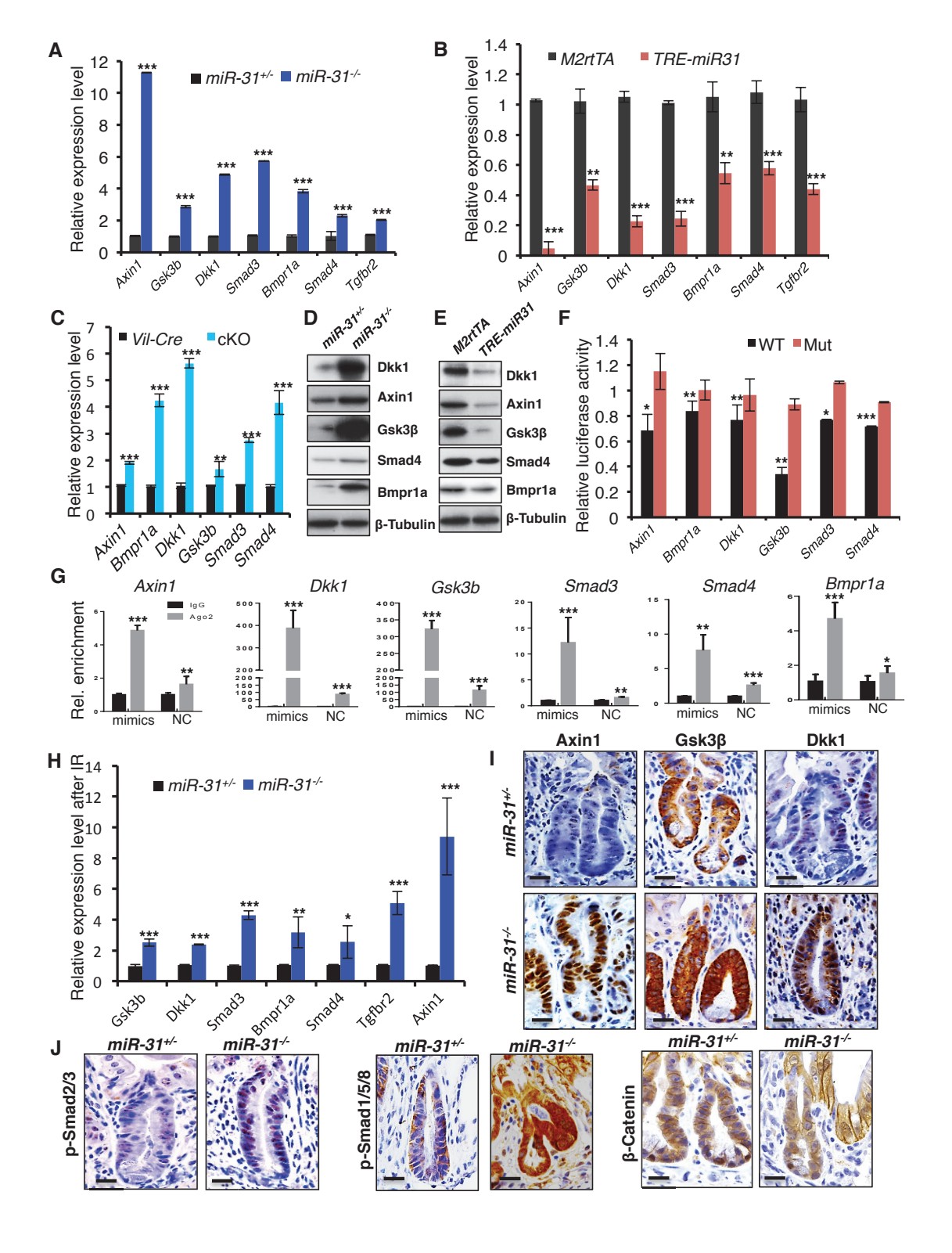

**Figure 6.** Identification of *miR-31* target genes in intestinal epithelium. (**A, B**) qRT-PCR analysis for *Axin1, Gsk3b, Dkk1, Smad3, Bmpr1a, Smad4* and *Tgfbr2* in *miR-31⁺/⁻* and *miR-31⁻/⁻* intestine (**A**), as well as *M2rtTA* and *TRE-miR31* intestine following 2 weeks of Dox induction (**B**). **p<0.01; ***p<0.001. (**C**) qRT-PCR analysis for *Axin1, Bmpr1a, Dkk1, Gsk3b, Smad3,* and *Smad4* in *Vil-Cre* and cKO intestine. **p<0.01; ***p<0.001. (**D**) Western blotting for Axin1, Gsk3β, Dkk1, Smad4, and Bmpr1a in *miR-31⁺/⁻* and *miR-31⁻/⁻* intestine. β-Tubulin was used as a loading control, which is identical

*Figure 6 continued on next page*

*Figure 6 continued*

with *Figure 5A*. n = 3 biological replicates. (**E**) Western blotting for Axin1, Gsk3β, Dkk1, Bmpr1a and Smad4 in *M2rtTA* and *TRE-miR31* intestine following 2 weeks of Dox induction. β-Tubulin was used as a loading control. n = 3 biological replicates. (**F**) Ratio of luciferase activity of *miR-31* mimics versus scramble RNA in wild type and mutant 3'UTR constructs based on 3 independent experiments. *p<0.05; **p<0.01; ***p<0.001. (**G**) RNA crosslinking, immunoprecipitation, and qRT-PCR (CLIP-PCR) assay for *Dkk1, Axin1, Gsk3b, Smad3, Smad4* and *Bmpr1a* upon Ago2 antibody immunoprecipitates in response to *miR-31* mimics and scramble RNA (NC). IgG was used as a negative control. (**H**) qRT-PCR analysis for *Axin1, Gsk3b, Dkk1, Smad3, Bmpr1a, Smad4* and*Tgfbr2* in *miR-31*$^{+/-}$ and *miR-31*$^{-/-}$ intestine 4 days post 12 Gy γ-IR. n = 3 biological replicates. *p<0.05; **p<0.01; ***p<0.001. (**I**) Immunohistochemistry for Axin1, Gsk3β and Dkk1 in *miR-31*$^{+/-}$ and *miR-31*$^{-/-}$ intestinal crypts 4 days post 12 Gy γ-IR. Scale bar: 25 μm. (**J**) Immunohistochemistry for p-Smad2/3, p-Smad1/5/8 and β-Catenin in *miR-31*$^{+/-}$ and *miR-31*$^{-/-}$ intestinal crypts 4 days post 12 Gy γ-IR. Scale bar: 25 μm.

DOI: https://doi.org/10.7554/eLife.29538.030

The following source data and figure supplements are available for figure 6:

**Source data 1.** Source data for *Figure 6*.
DOI: https://doi.org/10.7554/eLife.29538.034
**Figure supplement 1.** Identification of *miR-31* target genes.
DOI: https://doi.org/10.7554/eLife.29538.031
**Figure supplement 2.** Identification of *miR-31* target genes.
DOI: https://doi.org/10.7554/eLife.29538.032
**Figure supplement 3.** Identification of *miR-31* target genes.
DOI: https://doi.org/10.7554/eLife.29538.033

activity counterbalances Wnt signaling (*He et al., 2004*), our findings suggest that *miR-31* is an important amplifier of Wnt signaling during intestinal regeneration.

## *MiR-31* contributes to tumor growth through Wnt activation and TGFβ and BMP repression

Given that *miR-31* promotes proliferation and inhibits apoptosis in the ISCs, it is plausible that *miR-31* may function in intestinal tumorigenesis. Supporting this notion, *miR-31* has been found to be upregulated in human colorectal cancers and in colitis (*Bandrés et al., 2006*; *Cottonham et al., 2010*; *Wang et al., 2009*; *Yang et al., 2013*). We tested the role of *miR-31* in intestinal tumorigenesis and observed that *miR-31* mimics promoted proliferation of HCT116, SW480 and LOVO colon cancer cells in vitro (*Figure 7—figure supplement 1A*). Conversely, inhibition of *miR-31* with *anti-miR-31* abrogated growth of these cells (*Figure 7—figure supplement 1A*). We further performed xenograft assays using *miR-31* mimics- and inhibitor-treated HCT116 cells. Thirty days after grafting, tumor volume and weight were increased in *miR-31* mimic-treated tumors, and markedly reduced in *miR-31* knockdown tumors (*Figure 7A*). The decrease in tumor size from *miR-31* inhibition coincided with the reduction in Ki67$^+$ and Cyclin D1$^+$ proliferating cells (*Figure 7B* and *Figure 7—figure supplement 1B*), and correlated with reduced Wnt activity and increased BMP and TGFβ activities (*Figure 7—figure supplement 1B*). To verify these findings in more physiologically relevant settings, we examined tumor formation in the AOM-DSS (Azoxymethane-Dextran Sodium Sulfate) model of the inflammation-driven colorectal adenocarcinoma (*De Robertis et al., 2011*). In comparison with the controls, we observed a marked decrease in both tumor size and number in *miR-31*$^{-/-}$ mice (*Figure 7C*), along with a concomitant reduction in proliferating cells (*Figure 7D,E*), and reduced Wnt pathway and increased BMP and TGFβ activity (*Figure 7D,F*). This tumor-promoting effect of *miR-31* in mice became even more evident when *miR-31* was deleted in *Vil-Cre;Apc*$^{flox/+}$ mice. Intestinal adenomas form in this mouse model upon loss of heterozygosity at the *Apc* locus, which is relevant to human disease in that spontaneous loss of *Apc* is found in the vast majority of human colorectal cancer (*Kinzler et al., 1991*; *Nagase et al., 1992*). Loss of *miR-31* in this animal model remarkably reduced tumor burden (*Figure 7G*), which was associated with decreased Wnt activity, enhanced BMP and TGFβ signaling, and decreased proliferating cells (*Figure 7H–J* and *Figure 7—figure supplement 1C*). Correspondingly, the *miR-31* targets Axin1, Dkk1, Gsk3β, Smad4 and Bmpr1a were up-regulated in the *miR-31* null tumors (*Figure 7—figure supplement 1D*). Together, these data demonstrate that *miR-31* plays an oncogenic role in intestinal and colorectal tumorigenesis by mediating activation of Wnt and repression of BMP and TGFβ signaling pathways.

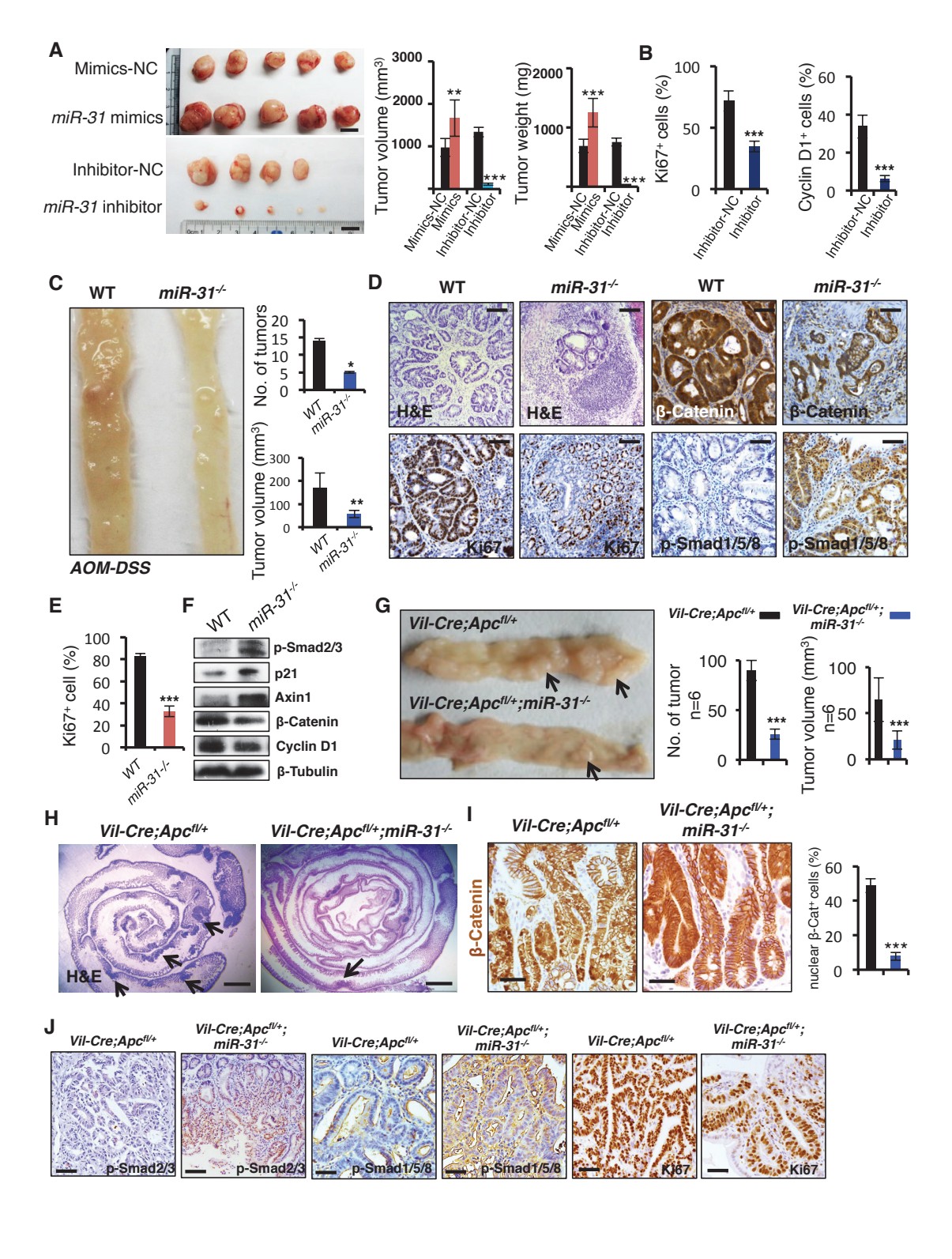

**Figure 7.** *MiR-31* promotes tumor growth in vivo. (**A**) Gross appearance of tumors of HCT116 colorectal cancer cell xenograft 30 days post transplantation. HCT116 colorectal cancer cells were transfected with mimics-NC or *miR-31* mimics, and inhibitor-NC or *anti-miR-31* (inhibitor) for 36 hrs before xenograft. NC-mimics, n = 5; *miR-31* mimics, n = 5; NC-inhibitor, n = 4; anti-miR-31, n = 5. Quantification of tumor volume and tumor weight at indicated conditions. **p<0.01; ***p<0.001. Scale bar: 1 cm. (**B**) Quantification of Ki67[+] and Cyclin D1[+] cells in NC-inhibitor and *miR-31* inhibitor treated
*Figure 7 continued on next page*

Figure 7 continued

tumors in **Figure 7—figure supplement 1B**. ***p<0.001. (C) Representative photograph of distal colon resected from WT and *miR-31$^{-/-}$* mice at the end of AOM-DSS protocol. Frequency and tumor size of inflammation-driven colorectal adenomas in mice treated with the AOM-DSS protocol, with or without *miR-31* deletion. n = 6 mice per group, *p<0.05; **p<0.01. (D) H&E, and immunohistochemistry for Ki67, β-Catenin and p-Smad1/5/8 in adenomas of WT and *miR-31$^{-/-}$* mice resulting from AOM-DSS treatment. Scale bar: 100 µm. (E) Quantification of Ki67$^+$ cells in Panel **D**. ***p<0.001. (F) Western blotting for p-Smad2/3, p21, Axin1, β-Catenin, Cyclin D1 in adenomas of WT and *miR-31$^{-/-}$* mice resulting from AOM-DSS treatment. β-Tubulin was used as a loading control. (G) Representative photograph of intestine resected from *Vil-Cre;Apc$^{fl/+}$* and *Vil-Cre;Apc$^{fl/+}$;miR-31$^{-/-}$* mice at 6 months of age. Arrows point to tumors. Quantification of tumor number and tumor volume in intestines from these mice. n = 6 biological replicates. ***p<0.001. (H) Representative histology of intestine resected from *Vil-Cre;Apc$^{fl/+}$* and *Vil-Cre;Apc$^{fl/+}$;miR-31$^{-/-}$* mice at 6 months of age. Arrows point to tumors. Scale bar: 2.5 mm. (I) Immunohistochemistry for β-Catenin and quantification of nuclear β-Catenin positive cells in *Vil-Cre;Apc$^{fl/+}$* and *Vil-Cre;Apc$^{fl/+}$;miR-31$^{-/-}$* tumors. (Black, *Vil-Cre;Apc$^{fl/+}$*; Blue, *Vil-Cre;Apc$^{fl/+}$;miR-31$^{-/-}$*). n = 6 biological replicates. Scale bar: 50 µm. ***p<0.001. (J) Immunohistochemistry for p-Smad2/3, p-Smad1/5/8 and Ki67 in *Vil-Cre;Apc$^{fl/+}$* and *Vil-Cre;Apc$^{fl/+}$;miR-31$^{-/-}$* tumors. Scale bar: 50 µm.

DOI: https://doi.org/10.7554/eLife.29538.035

The following source data and figure supplement are available for figure 7:

**Source data 1.** Source data for **Figure 7**.
DOI: https://doi.org/10.7554/eLife.29538.037
**Source data 2.** Source data for **Figure 7—figure supplement 1**.
DOI: https://doi.org/10.7554/eLife.29538.038
**Figure supplement 1.** *MiR-31* promotes tumor growth.
DOI: https://doi.org/10.7554/eLife.29538.036

## STAT3 signaling pathway mediated *miR-31* expression in response to irradiation

Lastly, we asked how radiation injury induces *miR-31* expression. We analyzed a 2 kb region upstream of the transcription start site of the *miR-31* gene locus for the potential binding sites of transcription factors using the JASPAR database and identified one STAT3 and two NF-κB binding sites (**Figure 8A**). Interestingly, the STAT3 and NF-κB signaling pathways were shown to be activated in response to γ-IR, evidenced by p-STAT3 and p65 levels, respectively (**Figure 8B,C**). The activation of the STAT3 pathway occurred mainly in the regenerative foci where *miR-31* is highly induced, while NF-κB was more prominently activated in villi where little *miR-31* is present and not in the regenerative foci (**Figure 8D**). This suggested a link between STAT3 activity and *miR-31* upon irradiation. To verify whether active STAT3 signaling could induce *miR-31* expression, mICc12 intestinal epithelial cells were treated with IL-6, a known activator of the STAT3 signaling. Indeed, *miR-31* expression was significantly induced upon IL-6 treatment (**Figure 8E**), concomitant with the activation of the STAT3 pathway (**Figure 8F**). In contrast, inhibition of STAT3 signaling with Stattic prominently dampened *miR-31* induction response to IL-6 treatment (**Figure 8G**), and reduced STAT3 signaling (**Figure 8H**). This inhibitory effect on *miR-31* expression was further validated using *Stat3* siRNA (**Figure 8I,J**). Importantly, *miR-31* was induced by IL-6 in the organoid cultures, indicating that this is an epithelial cell-autonomous mechanism (**Figure 8K**). Luciferase reporter assays reveal that IL-6 is able to induce its activity, while mutation of the p-STAT3 binding site blocked it (**Figure 8L**). Furthermore, Chromatin Immunoprecipitation (ChIP) assays show that p-STAT3 is recruited to its binding site on the *miR-31* promoter (**Figure 8M**). Thus, our data strongly suggest that STAT3 activity potentiates *miR-31* induction to promote crypt regeneration in response to radiation injury.

## Discussion

The intestinal epithelium is one of the most rapidly renewing tissues (*Leblond and Walker, 1956*). Those Lgr5$^+$ CBC stem cells residing at the base of crypts maintain the proliferative capacity necessary to meet this demands of high-turnover tissue, which is driven by activation of the canonical Wnt pathway, as well as repression of BMP signaling (*Li and Clevers, 2010*), (*Li et al., 2014*), (*Kosinski et al., 2007*). Wnt pathway activity and BMP inhibition are believed to be the niche for cycling CBCs. However, it is largely unknown how those Lgr5$^+$ CBCs integrate the signals of Wnt antagonists and activators of BMP and TGFβ. Here we show that the *miR-31* activates Wnt signaling by directly repressing a cohort of Wnt antagonists *Dkk1*, *Axin1* and *Gsk3b*, and represses BMP/

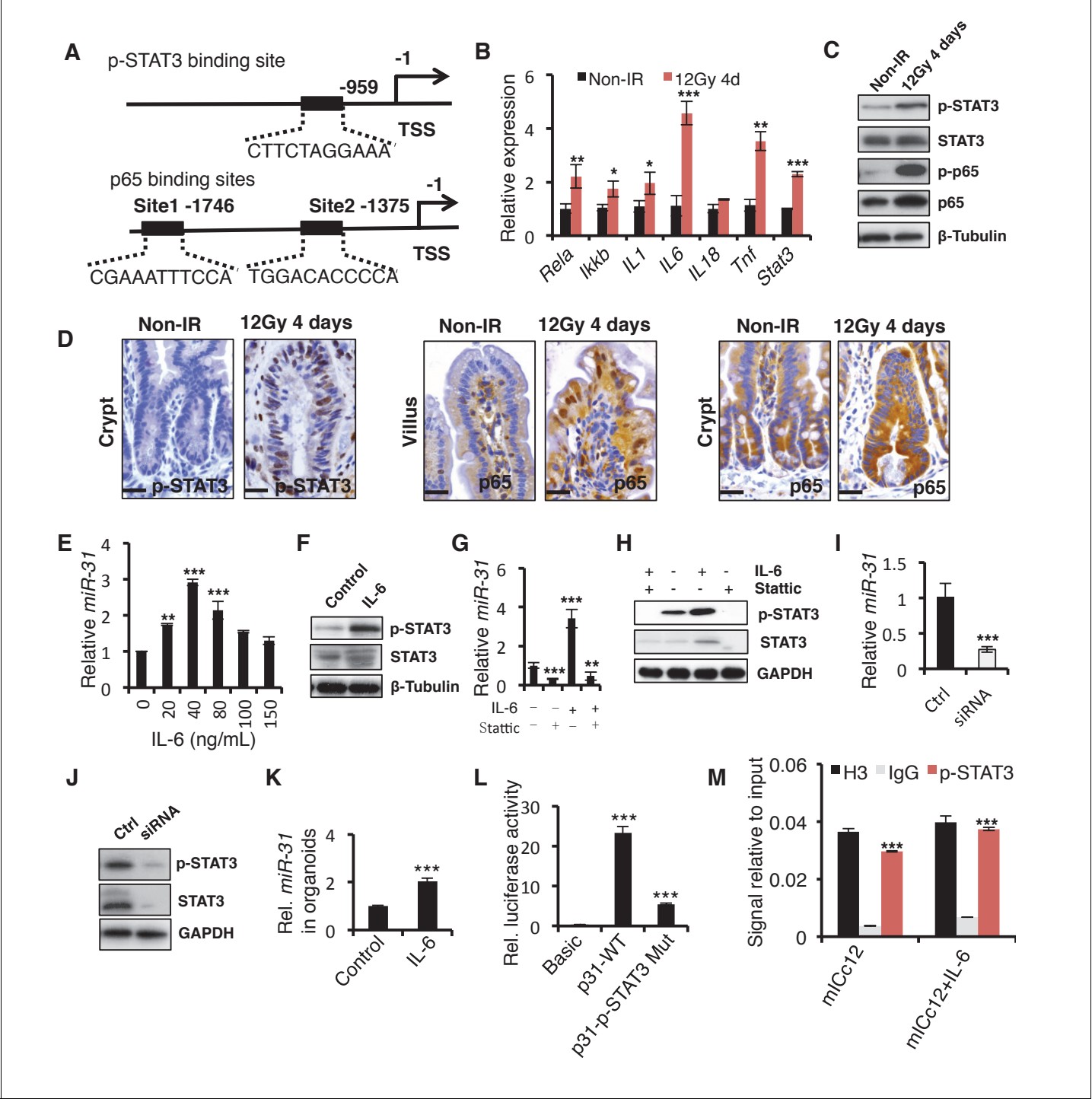

**Figure 8.** The STAT3 pathway mediates the induction of *miR-31* caused by γ-IR. (**A**) The schematic diagram showed two potential p65 binding sites and one p-STAT3 binding site in the *miR-31* promoter. (**B**) qRT-PCR analysis for *Rela*, *Ikk-b*, *IL-1*, *IL-6*, *IL-18*, *Tnf* and *Stat3* in the intestinal epithelium 4 days after exposure to 12 Gy γ-IR, relative to non-irradiated controls. n = 3 biological replicates. *p<0.05, **p<0.01, ***p<0.001. (**C**) Western blotting for STAT3, p-STAT3, p65 and p-p65 in the intestinal epithelium 4 days after exposure to 12 Gy γ-IR, relative to non-irradiated controls. n = 3 biological replicates. (**D**) Immunohistochemistry for p-STAT3 and p65 in control and the intestinal epithelium 4 days after exposure to 12 Gy γ-IR. n = 3 biological replicates. Scale bar: 25 μm. (**E**) qRT-PCR for *miR-31* in mouse intestinal epithelial cell line (mICc12) in response to IL-6 with concentrations of 20, 40, 80, 100 and 150 ng/mL. n = 3 technical replicates. **p<0.01; ***p<0.001. (**F**) Western blotting for STAT3 and p-STAT3 in mICc12 cells in response to 40 ng/mL IL-6. (**G**) qRT-PCR analysis for *miR-31* in mICc12 cells treated with IL-6 and STAT3 inhibitor, Stattic. **p<0.01; ***p<0.001. (**H**) Western blotting for p-STAT3 in mICc12 cells treated with IL-6 and Stattic. (**I**) qRT-PCR analysis for *miR-31* in mICc12 cells treated with *Stat3* siRNA. ***p<0.001. (**J**) Western

*Figure 8 continued on next page*

*Figure 8 continued*

blotting for STAT3 and p-STAT3 in mICc12 cells treated with STAT3 siRNA. (**K**) qRT-PCR analysis for *miR-31* in cultured organoids treated with IL-6.
n = 4 technical replicates. ***p<0.001. (**L**) Luciferase activity in lysates of mICc12 cells transfected with luciferase reporter plasmids of pGL3-basic empty
vector (basic), wild type *miR-31* promoter or mutant promoter with mutation of p-STAT3 binding sites. ***p<0.001. (**M**) Chromatin immunoprecipitation
(ChIP) assay carried out on mICc12 cells using antibodies against p-STAT3 and Histone 3. The antibody against Histone 3 was used as a positive
control. The enrichment of p-STAT3 binding to *miR-31* promoter was quantified using qPCR. ***p<0.001.
DOI: https://doi.org/10.7554/eLife.29538.039

The following source data is available for figure 8:

**Source data 1.** Source data for *Figure 8*.
DOI: https://doi.org/10.7554/eLife.29538.040

TGFβ signaling by directly inhibiting activators of the pathways, *Smad3*, *Smad4* and *Bmpr1a*, pointing to an important role of *miR-31* acting as a rheostat to integrating niche signals sensed by cycling CBCs. In agreement with this point, our in vivo analysis demonstrated that *miR-31* induction increases the number of Lgr5[+] CBCs whereas *miR-31* deletion reduces CBC frequency. Niche Wnt signals likely originate from sub-epithelial telocytes whose presence is required for CBC activity, and possibly to a lesser extent from Paneth cells, who secrete Wnt ligands but are dispensable for CBC activity (*Durand et al., 2012*; *Aoki et al., 2016*; *Sato et al., 2011*; *Kim et al., 2012*; *San Roman et al., 2014*; *Kabiri et al., 2014*). BMP antagonists noggin and gremlin are similarly secreted by sub-mucosal tissues below the crypts (*Kosinski et al., 2007*), repressing the BMP signaling in CBCs. Thus, sub-epithelial mesenchyme constitutes an extrinsic niche for cycling ISCs. In contrast to secretory signals from an extrinsic niche, *miR-31* appears to be an intrinsic coordinator of these extrinsic niche signals, supporting canonical Wnt and represses BMP/TGFβ signals within CBCs. Thus, we identify *miR-31* as a cell-autonomous post-transcriptional regulator of the ISC niche, maintaining proliferative capacity of cycling CBC cells. In addition, we also noticed that *miR-31* loss resulted in an increased apoptosis in CBC cells, suggesting the importance of *miR-31* in maintaining cell survival. The molecular mechanism by which *miR-31* protects against apoptosis warrants future study.

The response to high dose of γ-IR can be separated into two distinct stages. First, within 24 hours, the majority of CBCs die via apoptosis and subsequent mitotic death, caused by residual mis-repaired and unrepaired of DNA double-strand breaks (*Hua et al., 2012*). Next, between 24 hours and 4 days after γ-IR, rare surviving CBCs and quiescent reserve ISCs enter the cell cycle and form regenerative foci that produce mitotically active Lgr5[+] cells that repair lost epithelium (*Yousefi et al., 2016*; *Hua et al., 2012*). We assume that reserve ISCs also undergo the same process, although lack of direct evidence. In line with this, *miR-31* is dramatically reduced within the first 24 hours post γ-IR, most likely due to loss of CBCs. Loss of *miR-31* led to an marked increase in apoptosis in both CBCs and +4 cells 2 hours post-γ-IR. Based on our data, we conclude that during the first stage *miR-31* acts as an anti-apoptotic factor, protecting CBCs and reserve ISCs against apoptosis. During the second stage, the surviving stem cells start proliferating to repopulate the depleted intestinal epithelium. The surviving stem cells are relatively damage-resistant (*Tian et al., 2011*; *Takeda et al., 2011*; *Li et al., 2014*; *Yousefi et al., 2016*; *Ritsma et al., 2014*), a property attributed to their quiescence, a state likely maintained by BMP/TGFβ signaling and inactivation of Wnt signaling (*Li et al., 2014*; *Yousefi et al., 2016*; *He et al., 2004*). We show that *miR-31* is prominently induced at the regenerative foci 36 hr post-γ-IR and that *miR-31* activates Wnt, and represses BMP/TGFβ activities. This points to the potential importance of *miR-31* in activating the surviving ISCs. Given BMP/TGFβ inhibiting ability of *miR-31*, we speculate that the homeostatic insensitivity of reserve ISCs to Wnt ligands (*Yan et al., 2012*) results from their having active BMP and TGFβ pathways, that must be suppressed for cells to become competent to respond to Wnt ligands. Our findings suggest that *miR-31* functions as an activator of dormant reserve ISCs. We also want to mention that the expression patterns of *Bmi1* and *Hopx* are not specific to +4 position, as both of these transcripts are found non-specifically throughout the crypt base (*Li et al., 2014*; *Muñoz et al., 2012*; *Itzkovitz et al., 2011*). This means that *miR-31*-activated stem cells represent a complex population including +4 cells, surviving Lgr5[+] cells, and those TA cells dedifferentiated in response to irradiation. Taken together, our findings suggest that *miR-31* functions as the anti-apoptotic factor in ISCs during the early post-γ-IR stage, and, potentially, serves as the cell-intrinsic activator of surviving

ISCs regenerative foci promoting regeneration. Future studies will be needed to comprehensively test this idea.

Many reports have showed that *miR-31* is overexpressed in CRC tissues (*Bandrés et al., 2006*; *Cottonham et al., 2010*; *Wang et al., 2009*) and increases in progressively during progression from normal to inflammatory bowl disease (IBD) to IBD-related neoplasia (*Olaru et al., 2011*). We demonstrate that *miR-31* promotes tumor development using several models, including cancer cells xenografting, AOM- and DSS- induced inflammation-driven tumors, and *Apc*-loss driven tumors, characterized by activated Wnt, and repressed BMP/TGFβ signalings. Indeed, several reports showed that *miR-31* is overexpressed in colorectal cancer (CRC) tissues (*Bandrés et al., 2006*; *Cottonham et al., 2010*; *Wang et al., 2009*). Wnt signaling is aberrantly up-regulated in CRCs, which due primarily to mutations in the Wnt antagonist *APC* (*Novellasdemunt et al., 2015*). Our current study suggests that *miR-31* up-regulation might also contribute to Wnt activation in CRCs. In addition, decreased BMP and TGFβ signaling is also often found in CRCs (*Bellam and Pasche, 2010*; *Hardwick et al., 2008*), and can be the consequence of *miR-31* upregulation. As such, our data suggests that *miR-31* acts as the oncogenic microRNA in CRCs. Moreover, tight association between *miR-31* induction and STAT3 pathway activation in intestinal tissues is worth noting. Our molecular data suggest direct activation of *miR-31* expression by STAT3 signaling pathway. Indeed, many reports showed that constitutive activation of STAT3 is frequently detected in primary human colorectal carcinoma (*Kusaba et al., 2005*; *Corvinus et al., 2005*) and contributes to invasion, survival, and growth of colorectal cancer cells (*Tsareva et al., 2007*; *Lin et al., 2005*). Therefore, our current study suggests a signaling pathway involving STAT3, *miR-31* and WNT/BMP/TGFβ that promotes colorectal tumorigenesis.

In summary, we propose a model in which *miR-31* functions as a cell-intrinsic master modulator of the intestinal stem cell niche signaling during normal homeostasis, regeneration and tumorigenesis (*Figure 9*). During homeostasis, *miR-31* functions to integrate niche signals, supporting canonical Wnt activity and represses BMP/TGFβ signaling pathways within cycling CBC stem cells. *MiR-31* is stress inducible and plays an important role in epithelial regeneration. In response to high dose of γ-IR, *miR-31* is markedly induced via STAT3 signaling pathway, and appears capable of regulating the

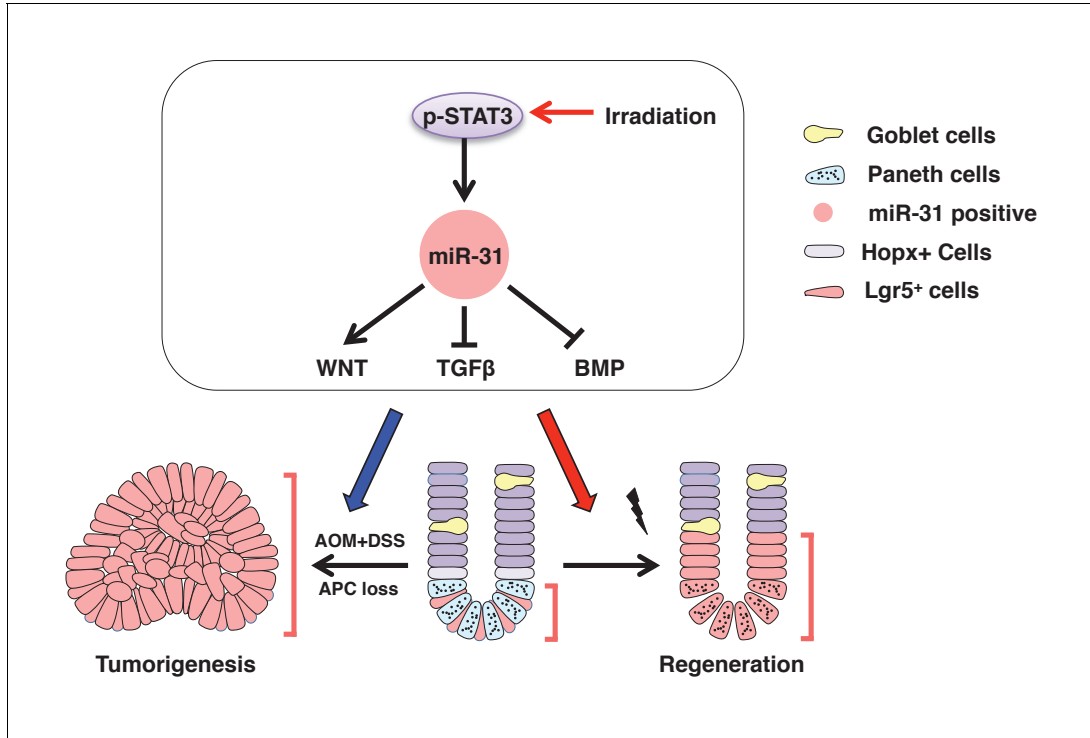

**Figure 9.** The *miR-31* working model in intestinal epithelial regeneration and tumorigenesis.
DOI: https://doi.org/10.7554/eLife.29538.041

activation state of a population of dormant, radiation resistant reserve ISCs during regeneration. Further, we demonstrate that *miR-31* acts as an oncomiR in promoting tumor growth.

## Materials and methods

### Animal experiments

All mouse experiment procedures and protocols were evaluated and authorized by the Regulations of Beijing Laboratory Animal Management and strictly followed the guidelines under the Institutional Animal Care and Use Committee of China Agricultural University (approval number: SKLAB-2011-04-03).

### Mouse strains

To generate *TRE-miR-31* transgenic mice, the *mmu-miR-31* sequence was amplified using the following primers: Forward 5'-CTCGGATCCTGTGCATAACTGCCTTCA-3' (BamHI site was added), and Reverse 5'-CACAAGCTTGAAGTCAGGGCGAGACAGAC-3' (HindIII site was added), and was inserted into *pTRE2* vector (Clontech) to generate a *pTRE2-miR31* construct. *TRE-miR31* transgenic mice were produced using standard protocols and crossed with *Rosa26-rtTA* mice which harboring the modified reverse tetracycline transactivator (*M2rtTA*) targeted to and under transcriptional control of the Rosa26 locus. Constitutive *miR-31*$^{-/-}$ mice were generated using CRISPR/Cas9 approach at the Nanjing Animal Center, and 402 bp DNA fragment containing *miR-31* was deleted to produce the null allele. Conditional *miR-31* KO allele was generated at the Shanghai Model Animal Center, the first exon (14806–15522) of *miR-31* was targeted with flanking LoxP sites resulting in the 2 LoxP locus. *Villin-Cre* (*Vil-Cre*) mice were purchased from the National Resource Center of Model Mice (stock number:T000142). *mTmG*, *Lgr5-eGFP-CreERT*, *Apc* floxed, and *Rosa26-LSL-lacZ* mice were obtained from Jackson Laboratories (stock number: 007576, 008875, 009045 and 009427). *Hopx-CreERT* mice were obtained from John Epstein laboratory. *Axin2-LacZ* mice were obtained from Yi Zeng laboratory.

### Cell culture

HCT116, SW480 and LOVO human colorectal cancer cell lines are purchased from American Type Culture Collection (ATCC) and the mouse mICc12 intestinal epithelial cell line was obtained from the Institute of Interdisciplinary Research (Fudan University, Shanghai, China) who originally obtained them from Dr A Vandervalle (Institut National de la Santé et de la Recherche Médicale, Faculté X, Paris, France). They were confirmed to come from a mouse cell line by Beijing Microread Genetics Co., Ltd using STR profiling. No cell lines are on the list of commonly misidentified cell lines. We have tested for mycoplasma contamination using a Mycoplasma Detection Kit, and no mycoplasma contamination was detected in any of the cultures. These cell lines were cultured in *DMEM/F12* medium. The sequence of *miR-31* inhibitor is 5'-AGCUAUGCCAGCAUCUUGCCU-3'. The sequence of Scramble RNA is 5'-CAGUACUUUUGUGUAGUACAA-3'. The Sequence of *miR-31* mimics:

5'-AGGCAAGAUGCUGGCAUAGCU-3'
3'-CUAUGCCAGCAUCUUGCCUUU-5'
The sequence of negative control for *miR-31* mimics:
5'-UUCUCCGAACGUGUCACGUUU-3'
3'-ACGUGACACGUUCGGAGAAUU-5'.

### Doxycycline induction and isolation of intestinal epithelium

For the induction, 2 mg/mL Dox (Doxycycline hyclate, Sigma) was added to the drinking water along with 1% w/v sucrose. Mice were induced at 8 weeks of age. To isolate intestinal epithelial cells, mouse intestine was dissected longitudinally and rinsed three times with ice-cold 1x DPBS, then cut into 2–4 mm long pieces, incubated in 1x DPBS containing 2 mM EDTA and 0.2 mM DTT for 30 min at 4°C on a rotating platform. Suspended cells were then collected folowing gentle vortexing. To isolate intestinal crypts, rinsed small intestine was cut-opened and and villi were scraped using coverslip glass, the technique which left the crypts attached. Crypts were then detached after tissue incubation in 1x DPBS with 2 mM EDTA for 30 min at 4°C with gentle vortexing. Isolated crypts were counted and pelleted as previously described (*Sato et al., 2009*).

## Flow cytometry

Dissected intestine was incubated with 5 mM EDTA and 1.5 mM DTT in HBSS for 30 min at 4°C. Single cell suspension was produced following Dispase (BD Biosciences) treatment and passing cells through 40 µm cell strainer. Flow cytometry analysis was performed using BD LSR Fortessa cell analyzer (BD Biosciences). PI-negative cells were selected, then gated for single cells based on the forward-scatter height *vs.* forward-scatter width (FSC-H *vs.* FSC-W) and side-scatter height *vs.* side-scatter width (SSC-H *vs.* SSC-W) profiles. The size of the nozzle for all sorting runs was 100 µm (20 psi). Lgr5-eGFP$^+$ cells were quantified by flow cytometry in *TRE-miR31;Lgr5-eGFP-CreERT* and *M2rtTA;Lgr5-eGFP-CreERT* mice after two weeks of Dox treatment. Lgr5-eGFP$^+$ cells in *miR-31$^{+/-}$;Lgr5-eGFP-CreERT* and *miR-31$^{-/-}$;Lgr5-eGFP-CreERT* mice were quantified using the same method.

## Crypt organoid culture

Crypt culture was performed as previously described in Sato et al. (2009). A total of 500 isolated crypts from *TRE-miR31*, *M2rtTA*, *Vil-Cre* and *Vil-Cre;miR31$^{fl/fl}$* (cKO) mice were mixed with 80 µL of matrigel (BD Bioscience) and plated in 24-well plates. After matrigel polymerization, 500 µL of crypt culture medium *[advanced DMEM/F12 (Gibco), 2 mM Glutamax (Invitrogen), 100 U/mL penicillin, 100 µg/mL streptomycin (Sigma), 1 mM N-acetyl cysteine (Sigma), B27 supplement (Invitrogen), N2 supplement (Invitrogen), 50 ng/mL mouse, EGF (Peprotech), 100 ng/mL mouse Noggin (Peprotech) and 10% human R-spondin-1 (Peprotech)]* was added to *M2rtTA*, *Vil-Cre* and *Vil-Cre;miR-31$^{fl/fl}$* small intestine crypt cultures. For *TRE-miR31* culture, human R-spondin-1 was removed from the medium, and instead 2 µg/mL of Dox was added.

## Hybridization

For *miR-31 in situ* hybridizations, digoxigenin (DIG)-labeled probes (Exiqon) were used following the manufacturer's protocol. Both DIG-labeled *miR-31* and scrambled probes (Exiqon) were hybridized at 61°C. U6 probe was used as the positive control. In situ signals were detected by staining with Anti-DIG-AP antibody (Roche) and developed using BM purple substrate (Roche).

## Quantitative RT-PCR

Total RNA was isolated from total mouse small intestinal epithelial cells using TRIzol reagent (Life Technologies) according to the manufacturer's instructions. Each RNA sample was reverse transcribed with the M-MLV Reverse Transcriptase (Sigma) using Oligo (dT) primers. Real-time PCR was performed using the LightCycler 480 SYBR Green I master mix on a LightCycler 480 real-time PCR system (Roche). qRT-PCR primers were follows:

*Axin1*-forward: 5'- TTCTGGGTTGAGGAAGCAGC −3'; *Axin1*-reverse: 5'- GATTAGGGGCTGGATTGGGT-3';

*Axin2*-forward: 5'- GGCTAGCTGAGGTGTCGAAG −3'; *Axin2* -reverse: 5'- GCCAGTTTCTTTGGCTCTTT −3';

*Ctnnb1*-forward: 5'- TCCTAGCTCGGGATGTTCAC −3'; *Ctnnb1* -reverse: 5'- TTCTGCAGCTTCCTTGTCCT −3';

*Bmpr1a*-forward: 5'- GCTGTCATCATCTGTTGTCCTGG −3'; *Bmpr1a*-reverse: 5'- CATTACCACAAGGGCTACACCACC −3';

*Myc*-forward: 5'- CTACTCGTCGGAGGAAAG −3'; *Myc*-reverse: 5'- ACTAGACAGCATGGGTAAG −3';

*Ccnd1*-forward: 5'- TGGTGAACAAGCTCAAGTGG −3'; *Ccnd1*-reverse: 5'- GGCGGATTGGAAATGAACT −3';

*Dkk1*-forward: 5'- TCCGAGGAGAAATTGAGGAA −3'; *Dkk1*-reverse: 5'- CCTGAGGCACAGTCTGATGA −3';

*Gsk3b*-forward: 5'- CCAACAAGGGAGCAAATTAGAGA −3'; *Gsk3b*-reverse: 5'- GGTCCCGCAATTCATCGAAA −3';

*Id1*-forward: 5'- ACCCTGAACGGCGAGATC −3'; *Id1*-reverse: 5'- GCGGTAGTGTCTTTCCCAGA −3';

*Id2*-forward: 5'- CTACTCGTCGGAGGAAAG −3'; *Id2* -reverse: 5'- ACTAGACAGCATGGGTAAG −3';

*Id3*-forward: 5'- TCCGGAACTTGTGATCTCCA −3'; *Id3*-reverse: 5'- GTAAGTGAAGAGGGC TGGGT −3';

*Junb*-forward: 5'- CGGATGTGCACGAAAATGGA −3'; *Junb*-reverse: 5'- GACCCTTGA- GACCCCGATAG −3';

*Msx1*-forward: 5'- CAGAGTCCCCGCTTCTCC −3'; *Msx1*-reverse: 5'- CTGAGCGAGCTGGAGAA TTC −3';

*Msx2*-forward: 5'- TTCACCACATCCCAGCTTCT −3'; *Msx2*-reverse: 5'- TTCAGCTTTTCCAG TTCCGC −3';

*Cdkn2b*-forward: 5'- GCCCAATCCAGGTCATGATG −3'; *Cdkn2b*-reverse: 5'- TCACACACA TCCAGCCGC −3';

*Cdkn2a*-forward: 5'- AGAGCTAAATCCGGCCTCAG −3'; *Cdkn2a* -reverse: 5'- CTCCCTCCCTCC TTCTGCT −3';

*Cdkn1a*-forward: 5'- ATCACCAGGATTGGACATGG −3'; *Cdkn1a* -reverse: 5'- CGGTGTCAGAG TCTAGGGGA −3';

*Cdkn1b*-forward: 5'- GGGGAACCGTCTGAAACATT −3'; *Cdkn1b* -reverse: 5'- AGTGTCCAGGGA TGAGGAAG −3';

*Cdkn1c*-forward: 5'- GTTCTCCTGCGCAGTTCTCT −3'; *Cdkn1c* -reverse: 5'- GAGCTGAAGGAC- CAGCCTC −3';

*Smad3*-forward: 5'- ACAGGCGGCAGTAGATAACG −3'; *Smad3*-reverse: 5'- AACGTGAACAC- CAAGTGCAT −3';

*Smad4*-forward: 5'- GGCTGTCCTTCAAAGTCGTG −3'; *Smad4*-reverse: 5'- GGTTGTCTCACC TGGAATTGA −3';

*Tgfbr2*-forward: 5'- TTGTTGAGACATCAAAGCGG −3'; *Tgfbr2*-reverse: 5'- ATAAAATCGACA TGCCGTCC −3';

*Rela*-forward: 5'- agataccaccaagacccacc-3'; *Rela*-reverse: 5'- ggtgaccagggagattcgaa −3';

*Ikkb*-forward: 5'-agaagtacaccgtgaccgtt-3';*Ikkb*-reverse: 5'-gggaagggtagcgaacttga-3';

*IL-1*-forward: 5'- tacctgtgtctttcccgtgg-3'; *IL-1*-reverse: 5'- ttgttcatctcggagcctgt-3';

*IL-6*-forward: 5'- gccagagtccttcagagaga-3'; *IL-6*-reverse: 5'-ggtcttggtccttagccact-3';

*IL-18*-forward: 5'- gtctaccctctcctgtaagaaca-3'; *IL-18*-reverse: 5'- tggcaagcaagaaagtgtcc-3';

*Tnf*-forward: 5'- aatggcctccctctcatcag-3'; *Tnf*-reverse: 5'- cccttgaagagaacctggga-3';

*Stat3*-forward: 5'- tgacatggatctgacctcgg-3'; *Stat3*-reverse: 5'- tgcccagattgcccaaagat −3';

For quantification of microRNA expression, mature *miR-31* was quantified using TaqMan micro-RNA assays according to the manufacturer's instructions. U6 snRNA was used as the internal control (Applied Biosystems).

## Histology, immunofluoresence, and immunochemistry

Intestines were rinsed with 1x DPBS, fixed in 10% formalin, paraffin-embedded and sectioned at 5 µm. Sections were stained with hematoxylin and eosin (H&E). For immunohistochemistry, antigen retrieval was performed by heating slides in 0.01 M citrate buffer (pH 6.0) in a microwave. Sections were then immunostained using ABC peroxidase method (Vector labs) with diaminobenzidine (DAB) as the enzyme substrate and hematoxylin as the counterstain. For immunofluorescence staining, paraffin sections were microwave pretreated in 0.01 M citrate buffer (pH 6.0), and incubated with primary antibodies, then incubated with secondary antibodies (Invitrogen) and counterstained with DAPI in the mounting medium (Vector labs). The following antibodies were used: anti-Ki67 (1:150, Leica), anti-GFP (1:200, Abcam), anti-Axin1 (1:100, Cell Signaling), anti-Gsk3β (1:2000, Abcam), anti-Dkk1 (1:50, Santa Cruz), anti-β-Catenin (1:500, Sigma), anti-BrdU (1:50, Abcam), anti-cleaved Caspase3 (1:100, Cell Signaling), anti-p-Smad1/5/8 (1:200, Cell Signaling), anti-p-Smad2/3 (1:200, Cell Signaling), anti-CyclinD1 (1:50, Abcam), anti-p65 (1:400, Cell Signaling), anti-p-STAT3 (1:800, Cell Signaling).

## Dual luciferase activity assays

To generate reporter constructs for luciferase assays, 300–600 bp fragments in length containing predicted *miR-31* target site in the 3'UTRs of *Axin1, Dkk1, Bmpr1a, Gsk3b, Smad3* and *Smad4* were cloned into the psiCHECK-2 vector (Promega) between the XhoI and NotI sites immediately downstream of the *Renilla* luciferase gene. To generate reporters with mutant 3'UTRs, nucleotides in the

target site complementary to the sequence of the *miR-31* seed region sequence were mutated using QuikChange Site-Directed Mutagenesis kit according to the manufacturer's protocol (Stratagene).

293T cells were seeded in 96-well plates one day before transfection. 10 ng of each reporter construct was co-transfected with *miR-31* mimics or scramble RNA at a final concentration of 50 nM into 293T cells using Lipofectamine 2000 according to the manufacturer's protocol (Invitrogen). After 24 hr, firefly and renilla luciferase activities were measured with the Dual-Glo luciferase assay system according to the manufacturer's instructions (Promega) and then be calculated using this formula (WT-mimics/WT-mimics NC) /(MUT-mimics/MUT-mimic NC).

The primers used for amplifying 3'-UTRs of candidate target genes of miR-31 were as follows:

*Dkk1*-forward: 5'-GCGCTCGAGTGGGCTTGAATTTGGTAT-3'; *Dkk1*-reverse:5'-TTAGCGGCCGCGTCCCGACTATCCTGTGA-3';

*Smad3*-forward: 5'-CCGCTCGAGCACCACACCGAATGAATG-3'; *Smad3*-reverse: 5'-ATAAGAATGCGGCCGCTGGCAATCCTTTACCATAGC-3';

*Gsk3b*-forward: 5'-TTAGCGGCCGCTCAGTTTCACAGGGTTAT-3'; *Gsk3b*-reverse: 5'-GCGCTCGAGACAAAGGCATTCAAGTAG-3';

*Axin1*-forward: GCCTCGAGTCAGTCAGGTGGACAGCC; *Axin1*-reverse:TAGCGGCCGCACACG-GACACTTGGAAGG;

*Bmpr1a*-forward: GCCTCGAGAATTAAACAATTTTGAGGGAG; *Bmpr1a*-reverse: TTGCGGCCGCCTACAGTTACAAGGTGGAT;

*Smad4*-forward: 5'- TTACTCCTAGCAGCACCC −3'; *Smad4*-reverse: 5'-CAGTTGTCGTCTTCCCTC-3';

## Western blotting

For western blotting assay, intestinal epithelial tissues were lysed in lysis buffer (Beyotime, China) with 1% PMSF (Phenylmethylsulfonyl fluoride). After quantification using a BCA protein assay kit (Beyotime, China), 30 μg of total protein was separated by 10% SDS-PAGE under denaturing conditions and transferred to PVDF membranes (GE Healthcare). Membranes were blocked in 5% nonfat dry milk in incubation buffer and incubated with primary antibodies, followed by incubation with the secondary antibody and chemiluminescent detection system (Pierce). The primary antibodies were: anti-GAPDH (Sigma), anti-β-Tubulin (Sigma), anti-CyclinD1 (Santa Cruz), anti-c-Myc (Santa Cruz), anti-β-Catenin (Sigma), anti-Dkk1 (Santa Cruz), anti-Gsk3β (Abcam), anti-Axin1 (Cell Signaling), anti-p-Smad2/3 (Cell Signaling), anti-p21(Santa Cruz), anti-Smad4 (Santa Cruz), anti-p-Smad1/5/8 (Cell Signaling), anti-Bmpr1a (Abcam), anti-p65 (Cell Signaling), anti-STAT3 (Cell Signaling), anti-p-p65 (Cell Signaling), anti-p-STAT3 (Cell Signaling).

## Irradiation injury

For irradiation, 2-month-old adult mice were subjected to 12 Gy γ-IR and executed at appointed time.

## Establishment of the AOM-DSS mouse model

Seven week-old control and *miR-31*$^{-/-}$ mice were intraperitoneally injected with AOM (Sigma-Aldrich,) at 10 mg/kg body weight. One week after AOM injection, mice were treated with the so-called DSS cycle, comprised of two steps in which mice were fed with 2.5% (w/v) DSS (molecular weight 36,000–50,000, MP Biomedicals) for 7 days followed by 14 days of normal water feeding. Mice were subjected to a total of three DSS cycles. After treatment, mice were sacrificed and distal colon tissues were collected and tumor number and volume were evaluated.

## Luciferase assay for *miR-31* promoter activity

The transcript of primary *miR-31* is located at Chromosome 4, NC_000070.6 (88910557..88910662, complement) in the mouse genome. The upstream 2 kb region of transcript start site (TSS) was identified as the *miR-31* promoter in this study, which is located at Chromosome 4, NC_000070.6 (88910663..88912663) and was cloned into the pGL3-Basic reporter constructs. The binding site of STAT3 is located at 88911572–88911582. The binding site 1 and 2 of p65 are located at 88912038–88912048 and 88912409–88912419, respectively.

## Chromatin immunoprecipitation (ChIP) assay

ChIP assay was performed according to the manufacturer's protocol with minor modifications, using Simple-ChIP enzymatic chromatin immunoprecipitation kit (Cell Signaling Technology). The sonicated nuclear fractions were divided for input control and for overnight incubated at 4°C with p-STAT3 or the positive control with H3, negative control with IgG. The recruited genomic DNA from the ChIP assays was quantified by qPCR with primers specific to p-Stat3 binding elements of the *miR-31* promoter regions. Primers were as follows: p-STAT3-binding site forward: 5'-TCCAGG-CAAGAAAGTGAGGG −3'; *p*-STAT3- binding site reverse: 5'- TGAGTAACAGTGCAACAGAGC-3'.

## Apoptosis analysis

The 21nt oligonucleotide *miR-31* inhibitor (5-AGCUAUGCCAGCAUCUUGCCU-3) or negative control Scramble RNA (5-CAGUACUUUUGUGUAGUACAA-3) were transfected into HCT116 cells with or without CHIR99021 (GSK3β inhibitor). The apoptotic cells were evaluated by FITC-Annexin V/PI staining (BD PharMingen) and analyzed by FACS (Becton, Dickinson).

## RNA crosslinking, immunoprecipitation, and qRT-PCR (CLIP-PCR) assay

CLIP-PCR assay performed as previously described with modification (*Wang et al., 2015*). Cells were treated with scramble RNA or *miR-31* inhibitor, and then harvested after being irradiated at 400 mJ/cm$^2$ twice. They were then re-suspended in PXL buffer with RNAsin (Promega) and RQ1 DNAse (Promega), and spun at 15000 rpm for 30 min. Supernatant was collected. Protein A Dynabeads (Dynal, 100.02, Thermo Fisher) and goat anti-rabbit IgG (Jackson ImmunoResearch,) or Ago2 antibody were incubated for 4 hr at 4°C with rotation. The supernatant was added to the beads for 2–4 hr at 4°C. Beads were then washed twice and digested with Proteinase K (4 mg/ml) for 20 min at 37°C. RNA was then extracted using Trizol Reagent (Invitrogen) and quantified by qRT-PCR.

## Statistical analysis

All analyses were performed in triplicate or greater and the means obtained were used for independent t-tests. Asterisks denote statistical significance (*p<0.05; **p<0.01; ***p<0.001). All data are reported as mean ±SD. Means and standard deviations from at least three independent experiments are presented in all graphs.

## Acknowledgement

We are grateful to Bogi Andersen for editing the manuscript, and Yeguang Chen for providing the Apc floxed mice. ZY is supported by the National Natural Science Foundation of China (No. .81772984, 81572614, 31271584); Beijing Nature Foundation Grant (5162018); the Major Project for Cultivation Technology (2016ZX08008001, 2014ZX08008001); Basic Research Program (2015QC0104, 2015TC041, 2016SY001, 2016QC086); SKLB Open Grant (2015SKLB6-16). JS is supported by the National Natural Science Foundation of China (No. 31370830 and 11675134) and the 111 Project (No. B16029). MVP is supported by the NIH NIAMS grants R01-AR067273, R01-AR069653, and Pew Charitable Trust grant. TA is supported by the NIAMS/NIH grant R01 AR061474-01.

## Additional information

### Funding

| Funder | Grant reference number | Author |
|---|---|---|
| National Institutes of Health | R01 AR061474-01 | Thomas Andl |
| National Institutes of Health | R01-AR067273 | Maksim V Plikus |
| National Institutes of Health | R01-AR069653 | Maksim V Plikus |
| National Natural Science Foundation of China | 31370830 | Jinyue Sun Jianwei Shuai |

| National Natural Science Foundation of China | 11675134 | Jinyue Sun<br>Jianwei Shuai |
| National Natural Science Foundation of China | 81572614 | Zhengquan Yu |
| National Natural Science Foundation of China | 31271584 | Zhengquan Yu |
| National Natural Science Foundation of China | 81772984 | Zhengquan Yu |

The funders had no role in study design, data collection and interpretation, or the decision to submit the work for publication.

## Author contributions

Yuhua Tian, Data curation, Software, Formal analysis, Validation, Investigation, Methodology, Writing—original draft, Project administration; Xianghui Ma, Cong Lv, Data curation, Formal analysis, Validation, Investigation, Methodology; Xiaole Sheng, Yongli Song, Formal analysis, Methodology; Xiang Li, Formal analysis; Ran Zhao, Formal analysis, Investigation, Methodology; Thomas Andl, Data curation; Maksim V Plikus, Fazheng Ren, Christopher J Lengner, Wei Cui, Formal analysis, Writing—review and editing; Jinyue Sun, Investigation, Writing—review and editing; Jianwei Shuai, Investigation, Methodology, Writing—review and editing; Zhengquan Yu, Conceptualization, Resources, Data curation, Formal analysis, Supervision, Funding acquisition, Validation, Investigation, Writing—original draft, Project administration, Writing—review and editing

## Author ORCIDs

Maksim V Plikus http://orcid.org/0000-0002-8845-2559
Wei Cui http://orcid.org/0000-0003-2019-380X
Zhengquan Yu http://orcid.org/0000-0001-8696-2013

## Ethics

Animal experimentation: All mouse experiment procedures and protocols were evaluated and authorized by the Regulations of Beijing Laboratory Animal Management and strictly followed the guidelines under the Institutional Animal Care and Use Committee of China Agricultural University (approval number: SKLAB-2011-04-03).

## Decision letter and Author response

Decision letter https://doi.org/10.7554/eLife.29538.049
Author response https://doi.org/10.7554/eLife.29538.050

# Additional files

## Supplementary files

• Transparent reporting form
DOI: https://doi.org/10.7554/eLife.29538.042

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
