## [Decision Letter]

[Editors’ note: a previous version of this study was rejected after peer review, but the authors submitted for reconsideration. The first decision letter after peer review is shown below.]

Thank you for submitting your work entitled "Stress Responsive *miR-31* is the Master-modulator of Intestinal Stem Cells during Regeneration and Tumorigenesis" for consideration by *eLife*. Your article has been evaluated by Fiona Watt (Senior Editor) and three reviewers, one of whom is a member of our Board of Reviewing Editors. The reviewers have opted to remain anonymous.

Our decision has been reached after consultation between the reviewers. Based on these discussions and the individual reviews below, we regret to inform you that your work will not be considered further for publication in *eLife*.

As you will see, the reviewers find your work of interest but they raise substantial concerns as well. At this moment, these concerns preclude us from accepting the manuscript. If you think however that you can address the comments from the reviewers in a satisfactory and timely manner, we will consider a revised version for another round of reviews.

Specifically, we urge to use functional readouts rather than relying on changes in marker expression in analyzing the consequences of *miR-31* genetic manipulation on intestinal stem cells. This could possibly be done by in vivo lineage tracing assays and/or by quantified organoid assays. We also urge you to be specific on which part of the intestinal track you are examining: the small intestine, colon or both. As an additional point, you need to be more accurate in detailing the endogenous expression of the *miR-31* together with the short-term and long-term effects of modifying *miR-31* expression on the epithelium using lineage markers, IHC/in-situ hybridization.

In the individual reviews, you will find other detailed comments that when addressed should improve your manuscript significantly.

Reviewer #1:

The authors make the interesting observation that one microRNA, *miR-31* controls the fate of intestinal stem cells. *miR-31* does so during homeostasis, regeneration and in cancer. It targets both Wnt and BMP signaling, activating Wnt and repressing BMP. The function of *miR-31* is assessed by loss- and gain-of-function experiments in mutant mice.

Figure 1 – the authors heavily focus on the localization of *miR-31* expression in Lgr5^+^ cells, however they do not show whether this transcript is also expressed in other cell populations in the intestine under normal conditions. For example, is *miR-31* expressed in the +4 stem cell population under normal conditions? A higher magnification of the in situ in Figure 1—figure supplement 1 may help clarify this issue.

Figure 1 – the authors show a decrease in intestine Lgr5^+^ cells in *miR-31* deletion mouse model and an increase in the *miR-31* overexpression model. However, they do not fully describe the consequences of these changes in stem cell numbers under normal circumstances. Is intestine tissue homeostasis affected? Are there changes in other intestine cell populations (enterocytes, goblet cells, paneth cells, entero-endocrine cells)? The phenotypes of these models under are important to characterize before making conclusions about injury or tumorigenesis. Also, what happens to mice overexpressing *miR-31* for an extended amount time (beyond 2 weeks)? This question is also relevant to Figure 7.

Figure 4 – to demonstrate that *miR-31* induction is sufficient to maintain organoid growth in the absence of R-spondin, the authors should also perform a cell propagation assay over a period of a few passages.

Figure 7 – the authors claim that *miR-31* plays an oncogenic role in intestinal cancer, however they do not show what happens to the intestine when *miR-31*, itself, is overexpressed for an extended period of time? Is amplification of *miR-31*, itself, oncogenic or just leads to hyperproliferation? This is especially interesting to know if one where to do *miR-31* agonism experiments to induce regeneration during damage or restore barrier function.

Reviewer #2:

This manuscript by Tian et al. describes an analysis of the role of *miR-31* in physiological and stress-induced intestinal epithelial proliferation, uncovering a novel role for this miRNA as a cell-autonomous positive regulator of epithelial proliferation in this tissue. These effects correlate with induction of Wnt signaling and repression of TGFb/BMP signaling by *miR-31*. A few targets of *miR-31* that may explain these effects (negative regulators of Wnt signaling and components of the TGF/BMP pathways) were validated. *miR-31* is also shown to be a direct target transcript of STAT and NF-κB transcription factors, which can explain its upregulation in multiple models of intestinal injury. The strength of this study is the analysis of robust gain- and loss-of-function mouse models which convincing support the proposed role for *miR-31* in intestine. Despite the somewhat superficial mechanistic analyses (simply showing regulation of a few targets in the Wnt/TGFb/BMP pathways), the overall findings are significant and, in principle, appropriate for publication in *eLife*. However, many aspects of the study are sloppy or poorly described, so major revisions and additional experiments are needed prior to publication, as described in detail below.

1) The phrase "…*miR-31* is the master modulator of intestinal stem cells…" in the title is hyperbolic and should be toned down (perhaps, "a major modulator" would be more appropriate).

2) Figure 1—figure supplement 1: In situ hybridizations require negative controls (ideally *miR-31^-/-^* sections).

3) A better characterization of the *miR-31* transgenic and knockout alleles is needed. For example, in what tissues is the *miR-31* transgene expressed upon Dox administration? Are there any overt phenotypes associated with acute *miR-31* induction or germline *miR-31* deletion? Are *miR-31* knockout alleles transmitted at the expected Mendelian ratios? Expression in what tissues is shown in Figure 1—figure supplement 1?

4) The BrdU pulse experiments in Figure 1 should be followed out for 24, 48, and 72 hours which may reveal clearer differences between the genotypes. The images showing BrdU staining in *miR-31^+/-^* vs. *miR-31^-/-^* crypts (Figure 1) do not look overtly different.

5) The finding of reduced *Hopx*-derived crypts after injury (Figure 2) can be fully explained by the finding of increased apoptosis in +4 cells after irradiation (Figure 2). This should be acknowledged in the text.

6) Figure 1—figure supplement 2: Simple growth curves (i.e. cell counts over time) would be preferable to these Ki67 stains of cultured cells. Since these cells are uniformly proliferating, albeit possibly at different rates, they all should be Ki67 positive. In addition, a description of this cell line is needed in the Materials and methods (how was Dox-inducible *miR-31* established in these cells?). Also, what is the "control" condition used in this experiment (and Figure 4)? It is unlikely that the same control would be appropriate for anti-*miR-31* and Dox. The former needs a scrambled anti-*miR* and the latter needs untreated cells.

7) In multiple experiments, "scrambled RNA" is used as a negative control for anti-*miR-31* (e.g. Figure 5, Figure 7, etc.). The source of the anti-*miR-31* is not described (please add), but it is unlikely that a scrambled RNA molecule would be the appropriate control. Anti-*miRs* are usually modified oligonucleotides and a scrambled oligo of the same composition should be used as a negative control.

8) Figure 1—figure supplement 2: Showing an image of a single crypt is not sufficient to make the point that the stem cell compartment is expanded in *TRE-miR-31* colon. Quantification of numerous crypts in multiple mice is needed to make this point.

9) The histologic analysis shown in Figure 1—figure supplement 2 (crypt height, Ki67) should be shown in *Vil-Cre; miR-31^fl/fl^*mice as well.

10) Insufficient *Vil-Cre; miR-31^fl/fl^*mice are examined after irradiation or DSS (n=2 or n=3, respectively). Given the inherent variability of the effects of these treatments, analysis of more mice is needed. Weight loss and clinical scores should be examined in these mice after DSS so the phenotype can be more completely compared to the germline KO mice.

11) The ISH experiments in Figure 7—figure supplement 1 are inadequate to make the point that *miR-31* is upregulated in colon cancer (many samples would be needed with quantification). If this is previously established in the literature, I would recommend removing these data (or greatly expanding these analyses so the results are meaningful).

12) Figure 7: Quantification of tumor burden in these experiments needed. n=3 is a very small number for these types of experiments. The image quality is very poor in these figure panels.

13) Figure 8: How was the *miR-31* promoter defined (genomic coordinates, including coordinates of STAT, NF-κB binding sites)? What specifically was cloned to make the promoter reporter constructs? These plasmids are not described in the Materials and methods.

Reviewer #3:

Using a variety of KO/over-expression mouse models and in vitro culture systems, the authors investigate the role of *miR-31* in regulating stem cell-driven homeostasis, regeneration and cancer in the intestine. Evidence is presented for *miR-31* modulating Wnt, BMP and TGF-β signaling pathway activity to levels compatible with efficient homeostasis, regeneration and disease in vivo. These findings are proposed to identify *miR-31* as a valuable new therapeutic target for intestinal disease.

Overall, there are some interesting findings presented that clearly define an important role for *miR-31* in the intestine. However, the major claim that *miR-31* exerts its effects via specifically regulating stem cell activity in vivo is poorly substantiated.

1) The experiment data are mainly relate to the intestine but there is sporadic inclusion of data from the colon. Analyses should either be extended to include both organs or the focus restricted to only one of them to make it easier for the reader to follow.

2) In-situ hybridization panels for *miR-31* throughout the manuscript are difficult to interpret and need replacing with much higher resolution images to accurately document the location and relative levels of endogenous *miR-31* expression in the small intestine and colon. Is expression confined to the crypt base (stem cells + Paneth cells) or present throughout the TA compartment Perhaps RNAscope would help here? Intestinal tissue from the *miR-31* null mice should be included as a negative control.

3) The up-regulation of *miR-31* seems to occur at relatively late phase (4-5 days after damage-induced stress). Is there any reason that it does not express in acute phase despite it apparently being such a critical modulator of epithelial regeneration? Wouldn't you expect rapid induction of expression to effect epithelial repair and establish barrier function?

4) Figure 1 – where does *miR-31* expression accumulate in the intestinal epithelium following irradiation? Is it uniform across the epithelium in regenerating areas or is it restricted to certain lineages? Again, high quality in-situ hybridization analysis is needed here.

5) Figure 1—figure supplement 1 – why was analysis of the *miR-31* overexpression phenotype only conducted after 2 weeks? A detailed time-course with early time-points is needed to determine when the phenotype first presents and in which cell-type (using lineage markers) – is it selectively driven from the stem cell compartment? Any apoptosis apparent following gain/loss of *miR-31* expression? Any effect on the Paneth cells (important regulators of stem cell activity)?

6) What happens to the villi in the mice with hyperproliferative crypts? One would assume they increase in length if no additional apoptosis is present at the villus tips. Are the phenotypes observed following modulation of *miR-31* expression consistent throughout the small intestine? Does the length of the intestine/colon change in the various *miR-31* mouse strains?

7) Why was HCT116 chosen for the in vitro gain/loss of function experiments? Is the phenotype also consistent for other colon cancer cells harboring different mutation spectra (for example APC mutations in place of the β-catenin mutation)?

8) Figure 1 – the increased movement of crypt cells in the *miR-31* gain/loss experiments is not at all evident to me from the BrdU pulse-chase experiment figure panels. Entire crypt/villus units should be included here to document the more rapid emergence of BrdU-labelled cells onto the villi from the crypts. Do the villi get longer as a result of the enhanced cell migration or is there more apoptosis at the villus tips?

9) Figure 1 – the observed increase in Lgr5-driven GFP expression following overexpression of *miR-31* does not necessarily indicate increased stem cell numbers/activity as claimed. If *miR-31* is indeed a potent Wnt enhancer, then overexpression may simply be activating Lgr5 expression (as a Wnt target gene) on non-stem cells. To properly document an increase in Lgr5^+^ stem cell activity, in vivo lineage tracing must be performed. One would then expect an increase in the number of long-term tracing units observed or an increase in the organoid forming frequency of isolated crypts or sorted GFPhi cells.

The increased organoid budding depicted in Figure 1 is not very convincing – there appears to be a very modest increase in relation to the observed several fold-increase in putative Lgr5^+^ stem cells in vivo. A better way to functionally evaluate the proposed increase in Lgr5^+^ Stem cells would have been to sort for EGFPhi cells and to determine their organoid forming capacity in comparison to GFPhi cells isolated from crypts with endogenous *miR-31* expression levels.

Is the increase in apoptosis observed in the organoids following loss of *miR-31* expression evident in vivo?

10) What is the *miR-31* loss/gain phenotype in the colon? Any changes to the frequency of Lgr5^+^ cells with tracing/organoid forming capacity?

11) Figure 2 – does overexpression of *miR-31* increase the regeneration rate of the small intestine and colon following sub-lethal doses of γ-irradiation? Lgr5^+^ stem cells have actually been found to be quite radioresistant (Hua et al., Gastroenterology 2012) and Lgr5^+^ stem cells are indispensable for crypt regeneration following irradiation (Metcalfe et al., Cell Stem cell 2014) – these observations argue that Lgr5^+^ stem cells are driving crypt regeneration rather than reserve stem cell populations. Considering this, does Lgr5 expression change in response to *miR-31* gain/loss of expression following irradiation?

*Bmi1* has been shown to be expressed throughout the entire crypt (including stem cells) and *Hopx* expression is enriched in the Lgr5^+^ stem cell compartment (Munoz et al., EMBOJ 2012). This should at least be discussed/noted in the manuscript rather than referring to these markers as being irrefutable +4/reserve stem cell markers.

12) Figure 2 – it is very difficult to conclude anything from this *Hopx*-tracing experiment. What are the figure panels meant to be showing? All I can surmise from this is that *Hopx*-derived tracing from the crypt base is absent in *miR-31* null mice, which is to be expected from the increase in apoptosis observed at the crypt base (harboring the *Hopx^+^* cells) following irradiation. I don't see how you can conclude that *miR-31* is required for reserve stem cell activity. I think it far more likely that *miR-31* expression is a general proliferation driver proliferation within the crypt, including TA cells which can re-acquire stem cell functions to drive regeneration (see recent papers on plasticity observed within secretory/absorptive progenitors following damage).

Do *Hopx* expression levels change in response to modulation of *miR-31* expression levels? Accordingly, does *Hopx*-driven lineage tracing change?

13) Figure 3 – is colonic regeneration following DSS withdrawal accelerated in *miR-31* overexpressing mice?

14) Figure 1—figure supplement 2 – phenotype needs far better characterization, incorporating a proper time-course. What happens within the epithelium following *miR-31* deletion? Is the phenotype more general TA cell driven or does it originate from within the stem cell compartment? Apoptosis evident as observed with the *miR31* null mice?

15) Figure 4 – *Axin2* expression isn't completely abrogated following loss of *miR-31*, so it is not correct to claim that Wnt signaling is absent. Since *Axin2* expression encompasses both the stem cell and TA cell compartments in the crypt, the dramatic down-regulation throughout the crypt supports a more general effect on the entire proliferative crypt compartment rather than selectively on the stem cells.

I would like to see IHC for β-catenin performed on the intestines of the different *miR-31* mice lines (including an early time-point for the *miR-312* overexpression line) to document the effects on Wnt signaling status (including nuclear β-catenin) on the different crypt/villus compartments.

Figure 4/D – Although the effects on Wnt signaling are well documented, I would like to have seen a more unbiased approach towards deciphering the direct result of modulating *miR-31* expression in the intestine – comparative microarray expression profiling of WT vs. *mir-31* null vs. *mir31*-overexpressing crypts would likely have shed additional light onto the pathways being directly regulated by *miR-31*. Such analyses at early time-points would also have indicated which cell types are initially being affected. Better validation in the form of IHC/In-situ analysis of expression changes to Wnt pathway targets (and targets of the BMP/TGF-β pathways) would also have helped to clarify which cells are responding to the changes in *miR-31* expression within the crypts/villi. Are the target genes first changing within the stem cell compartment or is it a more general response? Are similar changes found in both the small intestine and colon?

Figure 4 – please accurately quantify the organoid data.

16) Figure 6 – again, a time-course would be helpful here to determine which genes are likely responding directly to *miR-31* expression changes. Since there are obviously working antibodies available for some of the target genes, it would also be nice to see IHC validation of changes occurring in the epithelium in the absence of irradiation. This would document where the changes are taking place within the intestinal epithelium. Are the findings also applicable to the colon?

17) Figure 7 – the tumor-suppressive effect of *miR-31* loss is clear, although not surprising given the proliferative block imposed on the cells. Does increased *miR-31* expression enhance tumor formation in this model? Given the increased expression of *miR-31* expression in human colon tumors, it would be interesting to see the effect of deleting *miR-31* in established mouse intestinal tumors – this would be more indicative of the therapeutic potential of blocking *miR-31* expression.

Figure 1 – again, why was HCT116 chosen for this experiment. Is the effect of *miR-3*1 expression modulation restricted to β-catenin mutants or also present in other mutant backgrounds (such as APC null)?

Figure 7 – the swiss role histology pictures appear to show a complete lack of tumors in the *miR-31* null mice, yet there are clearly tumors present on the whole-mount image of the intestine. A more representative picture should be used.

Why do some tumors still arise in an APC mutant background in *miR31* null mice? What is their Wnt status?

18) Is *miR-31* also up-regulated during epithelial regeneration in human intestine (organoids?)? I would like to know its expression status in inflammatory disease patients according to their therapeutic status. Is *miR-31* down-regulated when inflammation is well suppressed?

[Editors’ note: what now follows is the decision letter after the authors submitted for further consideration.]

Congratulations, we are pleased to inform you that your article, "Stress Responsive *miR-31* is a Major Modulator of Mouse Intestinal Stem Cells during Regeneration and Tumorigenesis", has been accepted for publication in *eLife*.

The authors report on the function of a microRNA, *miR-31* as controlling the fate of intestinal stem cells. *miR-31* can function as such during homeostasis, regeneration and in cancer. They find that *miR-31* targets both the Wnt and BMP signaling pathways, activating Wnt and repressing BMP. The function of *miR-31* is assessed by loss- and gain-of- function experiments in mutant mice.

If you have selected our "Publish on Acceptance" option, your PDF will be published within a few days; if you have opted out of the "Publish on Acceptance" option, your work will be published in about four weeks' time. Please take note of the points below and we hope you will continue to support *eLife* going forwards.

After an initial round of reviewing, the authors have come back with a revised version that is satisfactory to two of the first three reviewers. The other original reviewer was unable to review the revision but the editor feels that the paper passes the criteria to be accepted.

Reviewer #1:

The authors have done a good job in replying to the concerns I raised during the first round of reviewing. I recommend accepting the paper in its current form.

Reviewer #2:

The authors have substantially revised the manuscript in response to the extensive reviewer critique. The vast majority of my concerns have been adequately addressed, although I am somewhat disappointed that they failed to directly assay the proposed increase in Lgr5^+^ stem cells with the best functional assay available – namely the organoid formation assay. I do not understand why they were unable to determine the organoid forming efficacy of sorted Lgr5-GFP+ cells in their mouse model – this is a well-established model, which would have directly proven an increase in true Lgr5^+^ stem cells rather than simply an increase in Lgr5-driven GFP expression in non-stem cells.

---

## [Author Response]

[Editors’ note: the author responses to the first round of peer review follow.]

As you will see, the reviewers find your work of interest but they raise substantial concerns as well. At this moment, these concerns preclude us from accepting the manuscript. If you think however that you can address the comments from the reviewers in a satisfactory and timely manner, we will consider a revised version for another round of reviews.

*Specifically, we urge to use functional readouts rather than relying on changes in marker expression in analyzing the consequences of miR-31 genetic manipulation on intestinal stem cells. This could possibly be done by* in vivo *lineage tracing assays and/or by quantified organoid assays.*

As the editors suggested, we first performed new organoid culture assays with control and *TRE-miR31 (miR-31* overexpressing) crypts. Both control and *TRE-miR31* mice were pretreated with Dox for two weeks. Compared to the controls, *TRE-miR31* crypts gave rise to more budding organoids (Figure 2), suggesting an increase of intestinal stem cells (ISCs) in *TRE-miR31* crypts. In our original manuscript, we showed that deleting *miR-31* within the intestinal epithelium results in a higher frequency of apoptotic organoids and compromised budding (Figure 2). Together, our findings strongly support the notion that *miR31* functionally promotes expansion of the ISCs.

Furthermore, we took advantage of *Lgr5-eGFP-CreERT;R26-LSL-LacZ* mice to perform lineage-tracing assay following 2, 4 and 7 days of *miR-31* induction. It revealed that *miR-31* overexpression increases the height of LacZ^+^ units at different time points, suggesting it promotes generation of new lineages from Lgr5^+^ ISCs (Figure 2 and Figure 2—figure supplement 1), From this we conclude that *miR-31* increases Lgr5^+^ stem cell numbers. We also noticed that the number of individual LacZ^+^ units derived from Lgr5^+^ stem cells is not significantly altered in *miR-31* overexpressing intestine, as compared to controls (Figure 2). We believe that this observation can be explained by the mosaic and inefficient activation of Cre in the inducible *Lgr5-eGFP-CreERT;R26-LSLLacZ* mouse model and that *miR-31* overexpression does not alter its efficiency.

We also urge you to be specific on which part of the intestinal track you are examining: the small intestine, colon or both.

This is a great suggestion. For this revision, we decided to focus the manuscript on the role of *miR-31* in small intestine during homeostasis, epithelial regeneration and tumorigenesis. Due to many differences between small intestine and colon, the DSS-treated colon data, while important, likely requires further in-depth investigation. Thus, in order to keep the manuscript focused on the core set of findings in the small intestine, for this revision we opted to remove the colon data.

As an additional point, you need to be more accurate in detailing the endogenous expression of the miR-31.

To address this criticism, we performed cell sorting to isolate Lgr5-GFP^high^, Lgr5-GFP^low^, Lgr5-GFP^neg^ cell subpopulations using *Lgr5-eGFP-CreERT* mice, and also isolate Hopx^+^ and Hopx^-^ cells, which is based on *Hopx-CreERT;mTmG* mice 15 hours after Tamoxifen injection. We quantified *miR-31* expression levels in isolated cells using qRT-PCR and found that *miR-31* expression levels are the highest in the Lgr5-GFP^high^ cell population with high progenitor potential, intermediate in Lgr5-GFP^low^ transient amplifying progenitor cells and the lowest in Lgr5-GFP^neg^ populations (Figure 1). Furthermore, higher levels of *miR-31* were also found in Hopx^+^ reserve stem cells as compared to Hopx^-^ epithelial cells (Figure 1). Furthermore, we performed *miR-31* in situhybridization. It shows that *miR-31* expression levels are generally higher in the crypts than in the villi (Figure 1). Together with the qRT-PCR results, these findings suggest that *miR-31* is broadly expressed in epithelial cells of the intestinal crypt and peaks inLgr5^high^ ISCs.

Together with the short-term and long-term effects of modifying miR-31 expression on the epithelium using lineage markers, IHC/in-situ hybridization.

To address this criticism, we examined the effect of *miR-31* overexpression on different types of differentiated cells following 2-weeks (short-term) and 1-year (long-term) of Dox induction, respectively. Dox treatment of *TRE-miR31* mice for 2 weeks resulted in (i) proliferative expansion of intestinal crypts (Figure 1 and Figure 1—figure supplement 3), (ii) reduction of differentiated cells such as enteroendocrine, goblet and paneth cells (Figure 1—figure supplement 4), and (iii) increase in apoptotic cells at the tip of the *TRE-miR31* villi (Figure 1). Surprisingly, sustained *miR-31* overexpression for 1 year resulted in very similar phenotypes as the 2-week induction (Figure 1—figure supplement 2 and Figure 1—figure supplement 3 and Figure 1—figure supplement 4). Thus, we conclude that the long-term effects of *miR-31* overexpression are quite similar to its short-term effects.

Next, we examined the effect of *miR-31* deletion on intestinal development at two weeks, two and eight months of ages. At these different time points, *miR31* deletion also produces consistent intestinal phenotypes that can be summarized as follows:

i) Deletion of *miR-31* significantly reduces crypt height and causes fewer proliferative cells in the crypts (Figure 1 and Figure 1—figure supplement 5).

ii) Deletion of *miR-31* induces apoptosis throughout thecrypt-villus axis (Figure 1—figure supplement 5). At the same time, apoptotic cells were predominantly present at the tips of control villi and very rare apoptotic cells appear in the control crypt-villus axis (Figure 1—figure supplement 5).

iii) Deletion of *miR-31* increases the number of enteroendocrine and Paneth cells, without the significant effect on goblet cells (Figure 1—figure supplement 6).

Compared to the 2-week-old *miR-31* KO mice, 2- and 8-month-old mutant mice exhibit slightly stronger phenotypes in terms of the changes in crypt height and apoptotic cell levels (Figure 1—figure supplement 5). The mutant phenotypes are quite similar between 2- and 8-month-old animals (Figure 1—figure supplement 5 and Figure 1—figure supplement 6).

In the individual reviews, you will find other detailed comments that when addressed should improve your manuscript significantly.

Reviewer #1:

[…] Figure 1 – the authors heavily focus on the localization of miR-31 expression in Lgr5^+^ cells, however they do not show whether this transcript is also expressed in other cell populations in the intestine under normal conditions. For example, is miR-31 expressed in the +4 stem cell population under normal conditions? A higher magnification of the in situ in Figure 1—figure supplement 1 may help clarify this issue.

We appreciate the reviewer’s suggestions. This criticism is essentially the same as the Editor’s point above. Please see our detailed third response to the Editor above.

Figure 1 – the authors show a decrease in intestine Lgr5^+^ cells in miR-31 deletion mouse model and an increase in the miR-31 overexpression model. However, they do not fully describe the consequences of these changes in stem cell numbers under normal circumstances. Is intestine tissue homeostasis affected? Are there changes in other intestine cell populations (enterocytes, goblet cells, paneth cells, entero-endocrine cells)? The phenotypes of these models under are important to characterize before making conclusions about injury or tumorigenesis.

This is an important point! To address it, we first characterized the phenotypes of *miR-31* overexpressing transgenic mice. Our findings were summarized below:

i) Following 2-week Dox induction, *TRE-miR31* mice exhibited a significant reduction in body weight (Figure 1) and their intestines were moderately, but significantly shorter than in controls (*Figure 1).*

ii) *miR-31* induction resulted in expansion of intestinal crypts with more proliferative cells (*Figure 1*). Also, more apoptotic cells were detected at the tips of *TRE-miR31* villi (Figure 1), suggesting enhanced epithelial cell turnover. However, the length of villi was mildly reduced in *TRE-miR31* mice, and the length of crypt/villus was not significantly altered in *TRE-miR31* mice, as compared to controls (Figure 1—figure supplement 2).

iii) We pulse-chased epithelial cells along the crypt-villus axis after a single dose of BrdU. Consistently, upward movement of BrdU^+^ cells from crypt to villi was enhanced in *TRE-miR31* mice (Figure 1—figure supplement 8).

iv) Fewer goblet, enteroendocrine and paneth cells were found in intestines from *TRE-miR31* mice following 2-week Dox induction, suggesting impaired cell differentiation (Figure 1—figure supplement 4).

Together, these results suggest that *miR-31* induction accelerates the conveyer belt-like movement of ISC progeny into the villi and their shedding into the lumen, which could comprise the differentiation of specialized intestinal cell types.

Next, we examined the consequence of *miR-31* loss in both *miR-31* germline knockout (KO) and *Villin-Cre*-driven intestinal epithelial conditional KO (cKO) mice. We followed these mice up to six months. Both *miR-31* KO and cKO mice are viable and fertile with no apparent gross phenotypes observed. Upon closer examination of their intestines, we found that:

i) Deletion of *miR-31* leads to a significant reduction in crypt height with fewer proliferative cells (Figure 1 and Figure 1—figure supplement 5).

ii) Deletion of *miR-31* gives rise to apoptotic cells throughout thecrypt-villus axis. At the same time, apoptotic cells were predominantly present at the tips of control villi and very rare apoptotic cells appear in the control cryptvillus axis (Figure 1—figure supplement 5). Analogous phenotypes were also found in the cKO mice (Figure 1—figure supplement 7).

iii) Deletion of *miR-31* led to increased number of enteroendocrine and paneth cells, while the number of goblet cells is not altered in miR-31 KO intestines (Figure 1—figure supplement 6).

iv) In a pulse-chase experiment after a single dose of BrdU, upward movement of BrdU^+^ cells from crypts to villi was impaired in KOmice (Figure 1—figure supplement 8).

Also, what happens to mice overexpressing miR-31 for an extended amount time (beyond 2 weeks)? This question is also relevant to Figure 7.

To examine the long-term effect of *miR-31* induction, we treated control and *TRE-miR31* mice with Dox for 1 year. We found that long-term *miR-31* induction gives rise to proliferative expansion of crypts, increased apoptosis at

the tip of the villi and reduction in the differentiated cell numbers (Figure

1—figure supplement 2C, E and Figure 1—figure supplement 3 and Figure 1—figure supplement 4). This phenotype is quite similar to that of *TRE-miR31* mice induced for two weeks. This data suggests that sustained *miR-31* increases proliferation levels, but is not able on its own to induce tumor formation.

Figure 4 – to demonstrate that miR-31 induction is sufficient to maintain organoid growth in the absence of R-spondin, the authors should also perform a cell propagation assay over a period of a few passages.

Per reviewer’s suggestion, we performed cell propagation assay for five passages. We found that the Dox-treated and R-Spondin-free*TRE-miR31* organoids can be normally passaged for at least five times, which is similar to the control organoids cultured with R-Spondin. These results suggest that *miR-31* induction can compensate for R-Spondin and activate Wnt signaling pathway.

These data are now included in Figure 4.

Figure 7 – the authors claim that miR-31 plays an oncogenic role in intestinal cancer, however they do not show what happens to the intestine when miR-31, itself, is overexpressed for an extended period of time? Is amplification of miR-31, itself, oncogenic or just leads to hyperproliferation?

This is an important question, and it is the same as that above.

Please see our detailed third response to reviewer 1 above.

This is especially interesting to know if one where to do miR-31 agonism experiments to induce regeneration during damage or restore barrier function.

We agree with the reviewer that it will be very interesting to examine the function of *miR-31* agonism in barrier restoration. Histology and immunohistochemistry for Ki67 show that Dox-induced *miR-31* overexpression promotes intestinal epithelial regeneration after irradiation (Figure 3—figure supplement 1). Further, lineage-tracing assay shows that *miR-31* induction accelerates generation of new progeny from Lgr5^+^ ISCs in response to irradiation (Figure 3). These findings suggest that *miR-31* agonism is likely to accelerate regeneration and barrier restoration. To our knowledge, another effective approach for testing this idea would be to deliver *miR-31* mimics to injury area using nanoparticles. However, this technically challenging experiment lies out of the scope of the current study. We hope, future follow-up studies will investigate the role of *miR-31* in barrier restoration more comprehensively.

Reviewer #2:

[…] However, many aspects of the study are sloppy or poorly described, so major revisions and additional experiments are needed prior to publication, as described in detail below.

1) The phrase "…miR-31 is the master modulator of intestinal stem cells…" in the title is hyperbolic and should be toned down (perhaps, "a major modulator" would be more appropriate).

We agree with the reviewer. Following this suggestion, we changed the title to “Stress Responsive *miR-31* is a Major Modulator of Mouse Intestinal Stem

Cells during Regeneration and Tumorigenesis”.

2) Figure 1—figure supplement 1: In situ hybridizations require negative controls (ideally miR-31^-/-^ sections).

We thank the reviewer for pointing this out. In the revised manuscript, we used *miR-31* KO intestinal sections as the negative control. We also included *TRE-miR31 (miR-31* overexpressing) intestinal sections as the positive control.

These new data are now included in Figure 1.

3) A better characterization of the miR-31 transgenic and knockout alleles is needed. For example, in what tissues is the miR-31 transgene expressed upon Dox administration?

To determine the expression pattern of *miR-31* in *Rosa26-rtTA;TRE-miR31* mice,we performed in situ hybridization for *miR-31* and found it to be predominantly expressed in the intestinal epithelial cells, albeit *Rosa26* promoter is expected to be active in all tissues. This data is shown in Figure 1—figure supplement 1.

Are there any overt phenotypes associated with acute miR-31 induction or germline miR-31 deletion?

This question is nearly identical to a point by the first reviewer. Please see our second response to reviewer 1.

*Are miR-31 knockout alleles transmitted at the expected Mendelian ratios?*

We have performed statistic analysis and found the transmission of *miR31* knockout alleles generally meets the Mendelian ratios. This data is now included in Figure 1—figure supplement 5.

Expression in what tissues is shown in Figure 1—figure supplement 1?

Intestinal epithelial tissues were used to detect *miR-31* expression levels in both *TRE-miR31* and *miR-31* KO mice. We clarified it in the figure legends of the revised manuscript. These data are now included in Figure 1—figure supplement 1.

4) The BrdU pulse experiments in Figure 1 should be followed out for 24, 48, and 72 hours which may reveal clearer differences between the genotypes. The images showing BrdU staining in miR-31^+/-^ vs. miR-31^-/-^ crypts (Figure 1) do not look overtly different.

This is a great suggestion. For this revision, we analyzed the outcomes of pulse-chase experiments on epithelial cells in both *TRE-miR31* and *miR-31^-/-^* mice at 24, 48, and 72 hours following single dose of BrdU. We found that upward movement of BrdU^+^ cells from crypts to villi was enhanced in *TRE-miR31* mice, and impaired in *miR-31^-/-^* mice at all time points, which is consistent with our original results. These new data are now included in Figure 1—figure supplement 8.

5) The finding of reduced Hopx-derived crypts after injury (Figure 2) can be fully explained by the finding of increased apoptosis in +4 cells after irradiation (Figure 2). This should be acknowledged in the text.

We agree with the reviewer. We have now acknowledged this possibility in the text: “…*miR-31* deficiency-mediated reduction in proliferation and increase in apoptosis within both CBC and reserve ISC compartments could account for the impaired regeneration of miR-31 null intestine…”.

6) Figure 1—figure supplement 2: Simple growth curves (i.e. cell counts over time) would be preferable to these Ki67 stains of cultured cells. Since these cells are uniformly proliferating, albeit possibly at different rates, they all should be Ki67 positive.

This is a good suggestion. We agree with the reviewer that Ki67 immunofluorescence is not a highly definitive method for detecting proliferation of cultured cells. Instead, we have examined the effect of *miR-31* on proliferation of HCT116, SW480 and LOVO colorectal cancer cells using MTT assay. This new data has been shown in the Figure 7−figure supplement 1A.

In addition, a description of this cell line is needed in the Materials and methods (how was Dox-inducible miR-31 established in these cells?).

To make the Dox-inducible *miR-31* cells, HCT116 cells were cultured in DMEM medium until 90-95% confluence. 4 μg of pTet-On plasmid (Clontech) were transfected using Lipofectamine^TM^ 2000 according to the manufacturer’s instructions. Positive clones were selected following treatment with 800 μg/mL G418 for 2 weeks. Clones were seeded into 24-well plates with DMEM medium and genotyped by PCR. Positive clones were co-transfected with *TRE-miR31* plasmid and selection vector pTK-Hyg (Clontech) at a molar ratio of 20:1, and selected by addition of 600 μg/mL Hygromycin for 2 weeks. Surviving clones were PCR genotyped. The cells containing pTet-On and *TRE-miR31* plasmids were used to induce *miR-31* overexpression with Dox.

For this revision, instead of Dox-inducible *miR-31* cells, we chose to examine the effect of *miR-31* mimics on proliferation of HCT116 colorectal cancer cells using MTT assay in the revised manuscript. Thus, we have removed the data on Dox-induced *miR-31* overexpression in HCT116 cells, and we do not include this method in the revised manuscript.

Also, what is the "control" condition used in this experiment (and Figure 4)? It is unlikely that the same control would be appropriate for anti-miR-31 and Dox. The former needs a scrambled anti-miR and the latter needs untreated cells.

In the original manuscript, we used the scramble RNA as a control to both *anti-miR-31* and Dox induction. We agree with the reviewer that scramble RNA is not the most stringent control. In the revised manuscript, we repeated these assays using *miR-31* mimics instead of Dox-induced *miR-31* over-expression. We used *miR-31* inhibitor-NC and *miR-31* mimics-NC as the negative controls for *miR-31* inhibitor and mimics, respectively. Information of *miR-31* inhibitor-NC and *miR-31* mimics-NC is now included in the Materials and methods section. Thus, we removed old Dox-induction data and added new data in Figure 4.

7) In multiple experiments, "scrambled RNA" is used as a negative control for anti-miR-31 (e.g. Figure 5, Figure 7, etc.). The source of the anti-miR-31 is not described (please add), but it is unlikely that a scrambled RNA molecule would be the appropriate control. Anti-miRs are usually modified oligonucleotides and a scrambled oligo of the same composition should be used as a negative control.

The sequence of *miR-31* inhibitor is 5’-AGCUAUGCCAGCAUCUUGCCU3’, while the sequence of Scramble RNA is 5’-CAGUACUUUUGUGUAGUACAA-

3’. This information is now included under the Cell Culture section of the Materials and methods. Although the nucleotide composition of Scramble RNA is not exactly same as Anti-*miR-31*, they are quite similar. To our knowledge, this particular scramble RNA has been used as a common negative control to microRNA inhibitors in multiple previous studies (Qin et al., 2014; Zhang et al., 2016). Thus, we believe that this scramble RNA is acceptable as a validated negative control for *Anti-miR-31*.

8) Figure 1—figure supplement 2: Showing an image of a single crypt is not sufficient to make the point that the stem cell compartment is expanded in TRE-miR-31 colon. Quantification of numerous crypts in multiple mice is needed to make this point.

In response to suggestions from the editor and reviewer #3 we focused this revised manuscript solely on the role of *miR-31* in small intestine. Thus, we removed this piece of data. Generally, we agree with the reviewer that quantification of numerous crypts in multiple mice is necessary. We plan to consolidate this data in our future study, which will focus on the role of *miR-31* in colon.

9) The histologic analysis shown in Figure 1—figure supplement 2 (crypt height, Ki67) should be shown in Vil-Cre; miR-31^fl/fl^ mice as well.

In the revised manuscript, we included histological analysis on *Vil-Cre; miR-31^fl/fl^*(cKO)mice. We found that specific deletion of *miR-31* in the intestinal epithelium also led to a significant reduction in crypt height with fewer proliferative cells. This is consistent with the *miR-31* germline KO results. We included new cKO data in Figure 1—figure supplement 7.

10) Insufficient Vil-Cre; miR-31^fl/fl^ mice are examined after irradiation or DSS (n=2 or n=3, respectively). Given the inherent variability of the effects of these treatments, analysis of more mice is needed. Weight loss and clinical scores should be examined in these mice after DSS so the phenotype can be more completely compared to the germline KO mice.

We improved the robustness of our irradiation data by increasing the total number of *Vil-Cre* and cKO mice to n = 5 each.

Also, as mentioned above, all of the colon data has been removed in the revised manuscript. We will report DSS-treated colon data in the future follow-up study.

11) The ISH experiments in Figure 7—figure supplement 1 are inadequate to make the point that miR-31 is upregulated in colon cancer (many samples would be needed with quantification). If this is previously established in the literature, I would recommend removing these data (or greatly expanding these analyses so the results are meaningful).

This is a good suggestion! We have removed this piece of data from the revised manuscript. Instead, we now cite the following articles to demonstrate that *miR-31* is up-regulated in colorectal cancers.

References:

1) Bandres, E., Cubedo, E., Agirre, X., Malumbres, R., Zarate, R., Ramirez, N., Abajo, A., Navarro, A., Moreno, I., Monzo, M., et al. 2006. Identification by Real-time PCR of 13 mature microRNAs differentially expressed in colorectal cancer and non-tumoral tissues. Mol Cancer 5:29.

2) Cottonham, C.L., Kaneko, S., and Xu, L. 2010. *miR-21* and *miR-31* converge on TIAM1 to regulate migration and invasion of colon carcinoma cells. J Biol Chem 285:35293-35302.

3) Wang, C.J., Zhou, Z.G., Wang, L., Yang, L., Zhou, B., Gu, J., Chen, H.Y., and Sun, X.F. 2009. Clinicopathological significance of microRNA-31, -143 and -145 expression in colorectal cancer. Dis Markers 26:27-34.

4) Yang, M.H., Yu, J., Chen, N., Wang, X.Y., Liu, X.Y., Wang, S., and Ding, Y.Q. 2013. Elevated microRNA-31 expression regulates colorectal cancer progression by repressing its target gene SATB2. PLoS One 8:e85353.

12) Figure 7: Quantification of tumor burden in these experiments needed. n=3 is a very small number for these types of experiments. The image quality is very poor in these figure panels.

In the revised manuscript, 6 *Vil-Cre;APC^fl/+^* and 6 *Vil-Cre;APC^fl/+^;miR-31^-/-^* mice have been analyzed in the APC mutant tumor model experiment. We quantified the tumor burden in both *Vil-Cre;APC^fl/+^* and *Vil-Cre;APC^fl/+^;miR-31^-/-^* mice, and found that loss of *miR-31*led to its remarkable reduction in tumor burden. Further, we examined proliferation and apoptosis in *Vil-Cre;APC^fl/+^* and *Vil-Cre;APC^fl/+^;miR-31^-/-^* tumors. Consistent to the reduced tumor burden, we found a decrease in the number of proliferating cells and reduced numbers of nuclear β-catenin^+^ cells in *Vil-Cre;APC^fl/+^;miR-31^-/-^* tumors. These new data are now shown in Figure 7.

We have replaced the Figure 7 with higher resolution images. As they are low magnification images, it is hard to make them very clear. If needed, we could provide the original high-quality images.

13) Figure 8: How was the miR-31 promoter defined (genomic coordinates, including coordinates of STAT, NF-κB binding sites)? What specifically was cloned to make the promoter reporter constructs? These plasmids are not described in the Materials and methods.

The transcript of primary *miR-31* is located on Chromosome 4 at NC_000070.6 (88910557..88910662, complement) in the mouse genome. The 2 Kb region upstream of the transcription start site (TSS) was identified as the *miR31* promoter in this study. It is located on the Chromosome 4 at NC_000070.6 (88910663..88912663) and was cloned into the pGL3-Basic reporter constructs.

The binding site for STAT3 is located at 88911572-88911582. The p65 binding sites #1 and #2 are located at 88912038-88912048 and 88912409-88912419, respectively. This information is now included in the Materials and methods.

Reviewer #3:

[…] Overall, there are some interesting findings presented that clearly define an important role for miR-31 in the intestine. However, the major claim that miR-31 exerts its effects via specifically regulating stem cell activity in vivo is poorly substantiated.

1) The experiment data are mainly relate to the intestine but there is sporadic inclusion of data from the colon. Analyses should either be extended to include both organs or the focus restricted to only one of them to make it easier for the reader to follow.

We appreciate reviewer’s criticism. Following additional considerations, we made a strategic decision to solely focus our revised manuscript on the role of *miR-31* in small intestine during homeostasis, epithelial regeneration and tumorigenesis. Thus, all colon data has been removed from the revised manuscript.

2) In-situ hybridization panels for miR-31 throughout the manuscript are difficult to interpret and need replacing with much higher resolution images to accurately document the location and relative levels of endogenous miR-31 expression in the small intestine and colon. Is expression confined to the crypt base (stem cells + Paneth cells) or present throughout the TA compartment Perhaps RNAscope would help here? Intestinal tissue from the miR-31 null mice should be included as a negative control.

In the revised manuscript, we have included higher magnification images of *miR-31* in situ hybridization under both homeostatic and injury conditions (Figure 1). As the reviewer suggested, we have used *miR-31* KO intestinal sections as negative control. We also used *TRE-miR31 (miR-31* overexpressing) intestinal sections as positive control. The in situ assay reveals that *miR-31* expression levels are generally higher in the crypts than in the villi and that *miR-31* is predominantly expressed in the epithelial cells of intestinal crypt, including Lgr5^+^ and Hopx^+^ stem cells, and transit amplifying cells. These data is now included in Figure 1.

RNAScope is an excellent method for detecting RNA species at a higher resolution. We have consulted with the company producing these reagents. However, it does not work for mature microRNAs due to its short nucleotide sequence. We thus have to detect *miR-31* expression using traditional digoxigenin (DIG)-labeled probes (Exiqon) following the manufacturer’s protocol, as well as qRT-PCR.

3) The up-regulation of miR-31 seems to occur at relatively late phase (4-5 days after damage-induced stress). Is there any reason that it does not express in acute phase despite it apparently being such a critical modulator of epithelial regeneration? Wouldn't you expect rapid induction of expression to effect epithelial repair and establish barrier function?

In response to DSS treatment, *miR-31* level is continuously upregulated with time, while it is not significantly changed within the first 24 hours. In our original manuscript, we have shown that *miR-31* is regulated by both NF-κB and STAT3 signaling pathways, which are important inflammation signals. We believe that those inflammation signals are not activated within 24 hours after DSS treatment, resulting in no significant change of *miR-31* level. Considering the revised manuscript focuses on the role of *miR-31* in small intestine, we have removed the colon data.

Following 12 Gy γ-IR, *miR-31* levels transiently and markedly drop by 24 hours (full proliferative arrest/DNA damage response) (Figure 1), and then sharply upregulated 48 hours post-γ-IR (during initiation of regenerative proliferation from radioresistant ISCs), returning to baseline levels within one week (after full recovery) (Figure 1). We believe that reduction in *miR-31* levels during early phase results from severe loss of Lgr5^+^ cells. Indeed, most Lgr5^+^ cells, that highly express *miR-31*, were depleted within 24 hours following 12 Gy γ-IR. However, between 24 to 48 hours post-IR *miR-31* becomes markedly upregulated, and this coincides with the *miR-31* promoter activation by STAT3 signaling pathway (Figure 8). We hope that this clarifies the causes behind the observed *miR-31* dynamics.

4) Figure 1 – where does miR-31 expression accumulate in the intestinal epithelium following irradiation? Is it uniform across the epithelium in regenerating areas or is it restricted to certain lineages? Again, high quality in-situ hybridization analysis is needed here.

In situ hybridization data showed that *miR-31* is dramatically upregulated in the regenerative foci, but not uniformly so across the intestinal epithelium. New in situ images have been included in Figure 1 of the revised manuscript.

5) Figure 1—figure supplement 1 – why was analysis of the miR-31 overexpression phenotype only conducted after 2 weeks?

Indeed, we performed time-course histological analysis in both *M2rtTA* and *TRE-miR31* mice. It revealed that the heights of intestinal crypts in the control *M2rtTA* mice were not significantly altered at different time points in response to Dox treatment. In contrast, crypts were significantly expanded in *TRE-miR31* mice following 14 days of Dox treatment, and the height of expanded crypts stabilized and was maintained for up to 1 year upon continuous Dox induction (Figure 1—figure supplement 2). Given that crypt elongation reached it maximum levels within 2-weeks of Dox induction, we thus conducted most of the subsequent assays at this timepoint.

A detailed time-course with early time-points is needed to determine when the phenotype first presents and in which cell-type (using lineage markers) – is it selectively driven from the stem cell compartment?

To address this question, we characterized the phenotypes of *TRE-miR31* mice following 7, 10 and 14 days of Dox induction. No apparent phenotypes were found in *TRE-miR31* mice following 7-day Dox induction (Figure 1—figure supplement 2). The phenotype of crypt expansion started presenting in *TRE-miR31* mice after 10 days of Dox treatment (Figure 1—figure supplement 2). Next, we examined various differentiated cells using lineage markers and found that the numbers of Paneth cells in the crypts were prominently reduced, while goblet and enteroenderocrine cells reduced mildly, yet significantly in *TRE-miR31* villi (Figure 1—figure supplement 4). We also showed that the numbers of Lgr5^+^ intestinal stem cells increases in *TRE-miR31* mice following 10 days of Dox induction (Figure 1—figure supplement 3). Based on the above findings, we conclude that the effect of *miR-31* induction on intestine is extensive, and not restricted to Lgr5^+^ ISCs only.

Any apoptosis apparent following gain/loss of miR-31 expression?

To address this question, we performed immunohistochemistry for cleaved Caspase 3 in intestine from both *miR-31* KO and *TRE-miR31* mice. We found that loss of *miR-31* gave rise to a certain number of apoptotic cells throughout *miR-31* KO crypt-villus axis, while apoptotic cells are predominantly presented at the tip of control villi and very rare apoptotic cells are presented in crypt-villus axis (Figure 1—figure supplement 5). Consistently, we also found that conditional deletion of *miR-31* gave rise to a certain number of apoptotic cells in cKO crypt-villus axis, particular in the crypt region (Figure 1—figure supplement 7), while only rare cleaved-caspase3^+^ apoptotic cells were found in control crypts, and apoptotic cells was also mainly located at the tip of control villi. Morphometric analysis revealed that ectopic apoptosis was mainly restricted to the CBC cells of cKO crypts (Figure 2). By contrast, more apoptotic cells were found at the tip of *TRE-miR31* villi as compared to controls, while no significant change were found for apoptotic cells throughout crypt-villus axis between both control and *TRE-miR31* mice (Figure 1—figure supplement 3). Together, these findings suggest that *miR-31* loss induces ectopic apoptosis in the intestinal epithelium, particularly in the CBC cells.

Any effect on the Paneth cells (important regulators of stem cell activity)?

In the revised manuscript, we have performed immunofluorescence for Lysozyme that detects paneth cells in *TRE-miR31* mice. There are fewer paneth cells in *TRE-miR31* intestines, relative to controls (Figure 1—figure supplement 4). Conversely, more Paneth cells were found in *miR-31* KO intestines (Figure 1—figure supplement 6). Considering that the Paneth cells were thought to be the niche of Lgr5^+^ intestinal stem cells (Sato et al., 2011), the reduction of Paneth cells is contrary to the increased number of Lgr5^+^ cells in *TRE-miR31* mice. Thus, we do not believe that the expansion of intestinal stem cells in *TRE-miR31* mice is dependent on the niche signals from Paneth cells. This also brings up recent findings that conclusively show that Paneth cells are not actually important regulators of ISC activity or a critical source of Wnt ligands, as Paneth cell ablation, or ablation of Wnt production in all epithelial cells (via porcupine deletion) has no adverse effect on ISCs (Durand et al., 2012) (Kim et al., 2012) (Aoki et al., 2016) (San Roman et al., 2014).

*6) What happens to the villi in the mice with hyperproliferative crypts? One would assume they increase in length if no additional apoptosis is present at the villus tips.*

This is a good point. Immunohistochemistry assay for cleaved Caspase 3 showed more apoptotic cells at the tips of *TRE-miR31* villi (Figure 1 and Figure 1—figure supplement 3). Furthermore, we found that the length of villi is significantly, albeit moderately, shortened in *TRE-miR31* transgenic mice (Figure 1—figure supplement 2). At the same time, the total length of the crypt/villus units is not significantly altered (Figure 1—figure supplement 2). The findings suggest that *miR-31* induction promotes turnover of intestinal epithelial cells, but it does not significantly affect the length of the crypt/villus units.

Are the phenotypes observed following modulation of miR-31 expression consistent throughout the small intestine?

To address this question, we characterized histology of the duodenum, jejunum and Ileum in *TRE-miR31* mice treated with Dox for 2 weeks. The phenotype of crypt expansion in *TRE-miR31* mice is consistent in different intestinal segments. This new data is now included in Figure 1—figure supplement 2.

Does the length of the intestine/colon change in the various miR-31 mouse strains?

The intestine length in *TRE-miR31* mice was moderately but significantly shorter than in control animals following 2 weeks of Dox induction. This data is now shown in Figure 1.

No apparent differences in intestinal lengths was found between control and *miR-31* KO mice (Figure 1—figure supplement 5).

7) Why was HCT116 chosen for the in vitro gain/loss of function experiments?

We have documented the reasons of choosing HCT116 colorectal cancer cells below:

i) Firstly, we examined *miR-31* expression levels between several

colorectal cancer cell lines using qRT-PCR and found them to be the highest in HCT116 cells. qRT-PCR results are shown in Author response image 1.

**Author response image 1. respfig1:** qRT-PCR analysis for *miR-31* in HCT116, LOVO, COCA2, HT29 and SW480 colorectal cancer cells.

ii) Secondly, it is well known that HCT116 cells are able to form tumors

after xenografting. Thus, we selected HCT116 cells for their robust performance in the xenograft experiments.

iii) Thirdly, HCT116 cells are heterozygous for β-catenin, harboring one

wild type and one mutant allele (Ilyas et al., 1997). The mutation in HCT116 cells (deletion of codon 45) results in loss of highly conserved serine residues in the region of the protein that may be a target for the enzyme GSK3β. However, prior works showed that Wnt activity in HCT116 cells is regulated by both Gsk3β (Kaler et al., 2009), and Wnt3a (Wu et al., 2012) (Voloshanenko et al., 2013). Thus, although heterozygous for β-Catenin mutation, HCT116 cells are still responsive to Wnt signaling modulation, rather than having Wnt pathway constitutively and highly active.

For these three reasons, we believe that HCT116 cell line is suitable for studying *miR-31* functions.

Is the phenotype also consistent for other colon cancer cells harboring different mutation spectra (for example APC mutations in place of the β-catenin mutation)?

To address whether *miR-31* also functions in APC mutated cells, we examined the effect of *miR-31* mimics and the inhibitor on LOVO colorectal cancer cells. We show that *miR-31* mimics significantly promote proliferation of LOVO cells, and conversely *miR-31* inhibitor represses it (Figure 7—figure supplement 1). Thus, our findings suggest that *miR-31* has shared functions in both types of colorectal cancer cells, with either β-Catenin or APC mutations.

8) Figure 1 – the increased movement of crypt cells in the miR-31 gain/loss experiments is not at all evident to me from the BrdU pulse-chase experiment figure panels. Entire crypt/villus units should be included here to document the more rapid emergence of BrdU-labelled cells onto the villi from the crypts.

Per reviewer’s suggestion, the revised manuscript shows the entire crypt/villus units. We also performed BrdU pulse-chase experiment on epithelial cells in both *TRE-miR31* and *miR-31^-/-^* mice, tracing cells for 24, 48, and 72 hours after a single pulse of BrdU. We found that upward movement of BrdU^+^ cells from crypts to villi was enhanced in *TRE-miR31* mice, and impaired in *miR-31^-/-^* mice. This is consistent to our original results. This new data is now included in Figure 1—figure supplement 8.

Do the villi get longer as a result of the enhanced cell migration or is there more apoptosis at the villus tips?

We believe this question largely overlaps with the reviewer 3’s point 6. Please see our first response to point 6 above.

9) Figure 1 – the observed increase in Lgr5-driven GFP expression following overexpression of miR-31 does not necessarily indicate increased stem cell numbers/activity as claimed. If miR-31 is indeed a potent Wnt enhancer, then overexpression may simply be activating Lgr5 expression (as a Wnt target gene) on non-stem cells. To properly document an increase in Lgr5^+^ stem cell activity, in vivo lineage tracing must be performed. One would then expect an increase in the number of long-term tracing units observed or an increase in the organoid forming frequency of isolated crypts or sorted GFPhi cells.

This question overlaps with the Editor’s point. Please refer to our first response to Editor’s comments.

The increased organoid budding depicted in Figure 1 is not very convincing – there appears to be a very modest increase in relation to the observed several fold-increase in putative Lgr5^+^ stem cells in vivo.

To consolidate this data, we have carefully quantified the budding organoids from control and *TRE-miR31* crypts. We found that *miR-31* induction significantly increases the frequency of budding organoids (Figure 2), supporting the notion that *miR-31* overexpression increases the number of Lgr5^+^ stem cells. The quantification analysis on budding number and crypt length was based on 3 independent experiments and 10 organoids were quantified at each set of experiment at day 3, day 5 and day 7. We conclude that *miR-31* induction significantly, albeit moderately, increases bud number per organoid and leads to more elongated crypts. This data is now shown in Figure 2—figure supplement 1.

A better way to functionally evaluate the proposed increase in Lgr5^+^ Stem cells would have been to sort for EGFPhi cells and to determine their organoid forming capacity in comparison to GFPhi cells isolated from crypts with endogenous miR-31 expression levels.

We agree that this would be a good method for determining the organoid forming capacity of Lgr5-GFP^high^ cells. For this revision, we made several attempts to isolate and culture Lgr5-GFP^high^ cells; however, due to technical difficulties, we were not able to optimize this assay sufficiently enough so that it produces consistent and reproducible results.

Is the increase in apoptosis observed in the organoids following loss of miR-31 expression evident in vivo?

10) What is the miR-31 loss/gain phenotype in the colon? Any changes to the frequency of Lgr5^+^ cells with tracing/organoid forming capacity?

Considering that the revised manuscript is focused on the role of *miR-31* in small intestine during homeostasis, epithelial regeneration and tumorigenesis, we chose to remove all colon data. We plan to investigate the colonic phenotypes in *miR-31* gain- and loss-of function models in our future, follow-up study.

11) Figure 2 – does overexpression of miR-31 increase the regeneration rate of the small intestine and colon following sub-lethal doses of γ-irradiation?

To address this question, we examined histology and proliferation in control and *TRE-miR31* intestines two and four days after 6 Gy of γ-IR following 2-week Dox induction. Surprisingly, we found that *miR-31* induction did not significantly affect the regenerative rate of small intestine after 6 Gy irradiation.

This data is shown in Author response image 2.

**Author response image 2. respfig2:** H&E and immunohistochemistry for Ki67 in intestines from *M2rtTA* control and *TRE-miR31* mutant mice 2 and 4 days post 6 Gy irradiation.

Lgr5^+^ stem cells have actually been found to be quite radioresistant (Hua et al., Gastroenterology 2012) and Lgr5^+^ stem cells are indispensable for crypt regeneration following irradiation (Metcalfe et al., Cell Stem cell 2014) – these observations argue that Lgr5^+^ stem cells are driving crypt regeneration rather than reserve stem cell populations. Considering this, does Lgr5 expression change in response to miR-31 gain/loss of expression following irradiation?

To address this question, we have performed qRT-PCR analysis for *Lgr5* in intestines from *TRE-miR31* and *miR-31* KO mice at 2 hours, 2 days and 4 days post-irradiation. *Lgr5* expression levels were remarkably up-regulated in *TRE-miR31* mice at different time points. Notably, *Lgr5* increases over 100 folds 4 days post-irradiation. Conversely, *Lgr5* expression levels were prominently reduced in *miR-31* KO mice at different time points. The data is now included in Figure 3. The results support the notion that *miR-31* protects Lgr5^+^ cells against apoptosis at the early stage after irradiation and promotes regeneration of de novo Lgr5^+^ cells at the late stages post irradiation.

Bmi1 has been shown to be expressed throughout the entire crypt (including stem cells) and Hopx expression is enriched in the Lgr5^+^ stem cell compartment (Munoz et al., EMBOJ 2012). This should at least be discussed/noted in the manuscript rather than referring to these markers as being irrefutable +4/reserve stem cell markers.

We thank the reviewer for pointing this out. We discuss this point in the revised manuscript as follows:

“…We also want to mention that the expression patterns of *Bmi1* and *Hopx* are not specific to +4 position, as both of these transcripts are found nonspecifically throughout the crypt base (Itzkovitz et al., 2011; Li et al., 2014; Munoz et al., 2012). This means that *miR-31*-activated stem cells represent a complex population including +4 cells, surviving Lgr5^+^ cells, and those TA cells dedifferentiated in response to irradiation …”

12) Figure 2 – it is very difficult to conclude anything from this Hopx-tracing experiment. What are the figure panels meant to be showing? All I can surmise from this is that Hopx-derived tracing from the crypt base is absent in miR-31 null mice, which is to be expected from the increase in apoptosis observed at the crypt base (harboring the Hopx^+^ cells) following irradiation. I don't see how you can conclude that miR-31 is required for reserve stem cell activity. I think it far more likely that miR-31 expression is a general proliferation driver proliferation within the crypt, including TA cells which can re-acquire stem cell functions to drive regeneration (see recent papers on plasticity observed within secretory/absorptive progenitors following damage).

We agree with the reviewer that this conclusion is overstated. In the revised manuscript we changed it to say: *“miR-31* deficiency-mediated reduction in proliferation and increase in apoptosis within the ISC compartments could account for the impaired regeneration of *miR-31* null intestine.”

Do Hopx expression levels change in response to modulation of miR-31 expression levels? Accordingly, does Hopx-driven lineage tracing change?

To address this question, we examined *Hopx* expression in response to *miR-31* induction and *miR-31* deletion. During homeostasis, the expression levels of *Hopx* are down-regulated in *TRE-miR31* intestine following 2-week Dox treatment, and, conversely, up-regulated in cKO intestine (Figure 2). Further, we performed lineage-tracing experiments with *Hopx-CreERT;R26-LSL-LacZ* mice. We found that *miR-31* induction markedly represses generation of lineages from Hopx^+^ cells during homeostasis. Interestingly, this is opposite to the effect of *miR-31* induction on Lgr5^+^ cells. These new data are now included in Figure 2 and Figure 2—figure supplement 1.

13) Figure 3 – is colonic regeneration following DSS withdrawal accelerated in miR-31 overexpressing mice?

As mentioned above, we removed all colon data form the revised manuscript and focused it solely on the role of *miR-31* in small intestine during homeostasis, epithelial regeneration and tumorigenesis. Colonic phenotypes will be investigated in more details in our future study.

14) Figure 1—figure supplement 2 – phenotype needs far better characterization, incorporating a proper time-course. What happens within the epithelium following miR-31 deletion?

This question is very similar to the Editor’s comment. Please also see our last response to the Editor above.

To address this question, we have carefully characterized the intestinal phenotypes of *miR-31* KO mice at 2 weeks, 2 months and 8 months of ages. At these different time points, deletion of *miR-31* produces very similar phenotypes in the intestine as follows:

i) Deletion of *miR-31* led to a significant reduction in crypt height with fewer proliferative cells (Figure 1 and Figure 1—figure supplement 5).

ii) Deletion of *miR-31* gave rise to apoptotic cells throughout thecrypt-villus axis, while normally apoptotic cells are restricted to the tips of villi (Figure 1—figure supplement 5).

iii) Deletion of *miR-31* led to increased numbers of enteroendocrine and paneth cells, while the numbers of goblet cells were not significantly altered (Figure 1—figure supplement 6).

Compared to the 2-week-old mice, crypt height and apoptotic phenotypes become somewhat more severe at 2 and 8 months of age (Figure 1—figure supplement 5), and no significant differences were observed between 2- and 8-months-old animals (Figure 1—figure supplement 5 and Figure 1—figure supplement 6).

Is the phenotype more general TA cell driven or does it originate from within the stem cell compartment?

In this study, we showed that the frequency of actively proliferating Lgr5^+^ ISCs is lower in *miR-31^-/-^* mice compared to controls (Figure 2). We also showed that proliferation of Lgr5^+^ ISCs is markedly suppressed in *miR-31^-/-^* mice (Figure 2). Further, our results showed more apoptotic CBCs in cKOmice (Figure 2) and higher frequency of apoptotic organoids derived from cKO crypts

(Figure 2). These data strongly indicate that *miR-31* deletion inhibits proliferative expansion of Lgr5^+^ CBCs, and concomitantly induces their apoptosis, and that these are major contributors to *miR-31* KO phenotype. Admittedly, we cannot completely rule out that compromised transit amplifying cell proliferation also contributes to the phenotype.

Apoptosis evident as observed with the miR31 null mice?

We believe this question is similar to point 5 above. Please see our third response to reviewer 3’s point 5.

15) Figure 4 – Axin2 expression isn't completely abrogated following loss of miR-31, so it is not correct to claim that Wnt signaling is absent.

We agree with the reviewer. We changed the manuscript text to say: “Wnt pathway activity was absent from CBCs of *miR-31^-/-^*crypts, appearing only faintly above the crypt base in the early TA zone.”

Since Axin2 expression encompasses both the stem cell and TA cell compartments in the crypt, the dramatic down-regulation throughout the crypt supports a more general effect on the entire proliferative crypt compartment rather than selectively on the stem cells.

I would like to see IHC for β-catenin performed on the intestines of the different miR-31 mice lines (including an early time-point for the miR-312 overexpression line) to document the effects on Wnt signaling status (including nuclear β-catenin) on the different crypt/villus compartments.

As the reviewer suggested, we now include β-catenin immunohistochemistry results in the revised manuscript. The number of nuclear β-catenin^+^ cells increases in the *TRE-miR31* intestine following 2-week and 2-month Dox induction, while it is not significantly altered following 7-day Dox induction. Conversely, it decreases in the *miR-31* KO intestine at two and four months of age. These findings indicate that *miR-31* activates Wnt signaling pathway, which is consistent with the Wnt reporter *Axin2-LacZ* results. These new data are now shown in Figure 4—figure supplement 1.

Figure 4/D – Although the effects on Wnt signaling are well documented, I would like to have seen a more unbiased approach towards deciphering the direct result of modulating miR-31 expression in the intestine – comparative microarray expression profiling of WT vs. mir-31 null vs. mir31-overexpressing crypts would likely have shed additional light onto the pathways being directly regulated by miR-31. Such analyses at early time-points would also have indicated which cell types are initially being affected.

As the reviewer suggested, we performed microarray analysis on intestinal tissues from three controls and three *TRE-miR31* mice following 7-day Dox induction, when phenotype is still not apparent. Transcriptome data from one pair of mice was excluded from analysis due to its low quality. Thus, our analyses are based on two control and two *TRE-miR31* mice. The heatmap analysis reveals that *miR-31* induction drives rapid and robust change in gene expression (see Author response image 3). Interestingly, Gene ontology (GO) analysis on both the biological process and KEGG pathways reveals significant enrichment for “Wnt signaling pathway” (Author response image 3), which is consistent with our earlier findings. Other top GO categories of interest include: (i) “Basal cell carcinoma”, (ii) “Hedgehog (Hh) signaling pathway”, (iii) “α-Linolenic Acid metabolism”, (iv)” GnRH signaling pathway”, and (v) “pathways in cancer” (Author response image 3). Considering that data is based on just two pairs of samples, we opted not to include it in the manuscript. However, if the reviewer thinks that it is worth including it into the supplement, we will be happy to do so.

**Author response image 3. respfig3:** Transcriptome profiling of *miR-31* overexpressing intestines.

Better validation in the form of IHC/In-situ analysis of expression changes to Wnt pathway targets (and targets of the BMP/TGF-β pathways) would also have helped to clarify which cells are responding to the changes in miR-31 expression within the crypts/villi. Are the target genes first changing within the stem cell compartment or is it a more general response? Are similar changes found in both the small intestine and colon?

To validate the effect of *miR-31* on Wnt signaling pathway, we first performed immunohistochemistry for β-Catenin in intestines from *miR-31* KO mice at 2 and 4 months of ages. We found that less nuclear β-Catenin positive cells were found in *miR-31* KO intestinal crypts, as compared to controls (Figure 4—figure supplement 1). Next, we treated both control and *TRE-miR31* mice with Dox for 7, 14 days and 2 months. The numbers of nuclear β-Catenin positive cells are greater in *TRE-miR31* intestinal crypts than controls following 14-day and 2-month Dox induction, while no significant difference were found between control and *TRE-miR31* mice following 7-day Dox induction (Figure 4—figure supplement 1). Further, we also showed that much less nuclear β-Catenin positive cells were found in *miR-31* null APC mutant tumors, as compared to APC mutant tumors (Figure 7). Together, these data strongly indicate that *miR31* activates Wnt signaling pathway. Nuclear β-Catenin positive cells were not found in intestinal villi from both *miR-31* KO and *TRE-miR31* mice (Author response image 4). It suggests that the response of Wnt signaling upon the changes of *miR-31* occur within intestinal crypts.

To validate the effect of *miR-31* on BMP and TGFβ signaling pathways, we examined p-Smad2/3 and p-Smad1/5/8 with immunohistochemistry in *miR-31* KO and *TRE-miR31* mice. We found that signals of p-Smad2/3 and p-Smad1/5/8 were upregulated in *miR-31* KO intestinal crypts, while they are down-regulated in *TRE-miR31* intestinal crypts following 7- and 14-day Dox induction (Figure 5—figure supplement 1). It suggests that *miR-31* represses BMP and TGFβ signaling pathways. Interestingly, we found that signals of p-Smad1/5/8 was upregulated in *miR-31* KO intestinal villi, and down-regulated in *TRE-miR31* intestinal villi (Author response image 4). No p-Smad2/3 positive cells were found in intestinal villi from *M2rtTA* and *TRE-miR31* mice, as well as control and *miR-31* KO mice (Author response image 4). The results suggest that the response of BMP signaling upon the changes of *miR-31* occur in both intestinal crypts and villi, while TGFβ signaling only do within intestinal crypts.

To further confirm this idea, we performed the rescue experiment with in vitro organoid culture system. We found that increasing concentrations of the BMP inhibitor Noggin in organoid culture was able to rescue the budding defect in *miR-31* cKO organoidsin a dose-dependent manner (Figure 5). Together, our findings strongly indicated an inhibitory role of *miR-31* in BMP and TGFβ signaling pathways.

Considering that the revised manuscript is focused on the role of *miR-31* in small intestine, we thus do not examine them in colon.

**Author response image 4. respfig4:** Immunohistochemistry for β-Catenin (**A**), p-Smad1/5/8 (**B**), and p-Smad2/3 (**C**) in intestinal villi from *M2rtTA* and *TRE-miR31* mice, as well as *miR-31^+/-^* and *miR-31^-/-^* mice.

Figure 4 – please accurately quantify the organoid data.

The quantification results are now included in Figure 4 in the revised manuscript.

*16) Figure 6 – again, a time-course would be helpful here to determine which genes are likely responding directly to miR-31 expression changes. Since there are obviously working antibodies available for some of the target genes, it would also be nice to see IHC validation of changes occurring in the epithelium in the absence of irradiation. This would document where the changes are taking place within the intestinal epithelium. Are the findings also applicable to the colon?*

To address this question, we have carefully performed immunohistochemistry for *miR-31* target genes such as Dkk1, Axin1, Gsk3β and Bmpr1a in intestines from *TRE-miR31* mice following 7 and 14 days of Dox induction during homeostasis. No apparent phenotype is present in the *TRE-miR31* mice following 7-day Dox induction. However, we found that the target genes are prominently reduced in intestinal crypts of *TRE-miR31* mice at 7-day time point, and that these changes persist at 14 days (Figure 6—figure supplement 2). Conversely, these genes are generally upregulated in *miR31^-/-^* intestinal crypts (Figure 6—figure supplement 2). No apparent difference of these genes was found in intestinal villi from *M2rtTA* and *TRE-miR31* mice, as well as control and *miR-31* KO mice (Author response image 5). The results suggest that these target genes are primarily responded to *miR-31* expression changes within intestinal crypts, which is consistent to higher expression level of *miR-31* in crypts. Together, these findings support the notion that the above genes are the direct targets of *miR-31*. To further confirm this notion, we performed RNA crosslinking, immunoprecipitation, and RT-PCR (CLIP-PCR) assays with Ago2 antibodies. The results show that transcripts of *Axin1, Dkk1, Gsk3*β, *Smad3, Smad4* and *Bmpr1a* are highly enriched in Ago2 immunoprecipitates, and that increasing *miR-31* activity augments their enrichment (Figure 6), providing evidence that *miR-31* directly binds to these transcripts. Considering that the revised manuscript is focused on the role of *miR-31* in small intestine, we thus do not examine these *miR-31* targets in colon.

**Author response image 5. respfig5:** Immunohistochemistry for Dkk1, Axin1, Gsk3β and Bmpr1a in intestinal villi from *M2rtTA* and *TRE-miR31* mice, as well as *miR-31^+/-^* and *miR-31^-/-^* mice.

17) Figure 7 – the tumor-suppressive effect of miR-31 loss is clear, although not surprising given the proliferative block imposed on the cells. Does increased miR-31 expression enhance tumor formation in this model?

To address this question, we transfected HCT116 colorectal cancer cells with *miR-31* mimics for 36 hours and then performed xenograft assays. Thirty days after grafting, tumor volume and weight were significantly increased in *miR31* mimics-treated tumors. These data are now included in Figure 7.

Given the increased expression of miR-31 expression in human colon tumors, it would be interesting to see the effect of deleting miR-31 in established mouse intestinal tumors – this would be more indicative of the therapeutic potential of blocking miR-31 expression.

Intestinal adenomas form in this mouse model upon loss of heterozygosity at the *APC* locus, which is relevant to human disease in that spontaneous loss of *APC* is found in the vast majority of human colorectal cancers (Kinzler et al., 1991; Nagase et al., 1992). Thus, we deleted *miR-31* in *Villin-Cre;APC^fl/+^*mice in this study. Loss of *miR-31*in this animal model remarkably reduced tumor burden (Figure 7), accompanied with the decrease in cell proliferation (*Figure 7*). These data suggest a therapeutic potential for *miR-31* inhibitor in blocking human colon cancer development.

Figure 1 – again, why was HCT116 chosen for this experiment.

In our opinion, this question relates to the point 7. Please see our detailed response above.

Is the effect of miR-31 expression modulation restricted to Β-catenin mutants or also present in other mutant backgrounds (such as APC null)?

To test this idea, we examined the effect of *miR-31* mimics and inhibitor on proliferation of LOVO colorectal cancer cells containing APC mutation. We show that *miR-31* mimics promote proliferation, and, conversely, *miR-31* inhibitor represses it (Figure 7—figure supplement 1). Here we also show that *miR-31* deletion remarkably inhibits growth of APC null tumors in vivo(Figure 7). Thus, we believe that the effect of *miR-31* modulation is also present in the APC mutant background.

Figure 7 – the swiss role histology pictures appear to show a complete lack of tumors in the miR-31 null mice, yet there are clearly tumors present on the whole-mount image of the intestine. A more representative picture should be used.

A more representative picture has been included in the revised manuscript, which is now shown in Figure 7.

Why do some tumors still arise in an APC mutant background in miR31 null mice? What is their Wnt status?

To address this questions, we performed immunohistochemistry for β-Catenin to detect Wnt activity in *Vil-Cre;APC^fl/+^*and *Vil-Cre;APC^fl/+^;miR-31^-/-^*tumors.The numbers of nuclear β-Catenin^+^ cells were dramatically reduced in *Vil-Cre;APC^fl/+^;miR-31^-/-^*tumors as compared to *Vil-Cre;APC^fl/+^*tumors (Figure 7), suggesting reduced Wnt activity in *miR-31* null tumors. However, few nuclear β-Catenin^+^ cells were still present in *miR-31* null tumors, indicating that Wnt activity is not completely repressed. We speculate that these Wnt-active cells could give rise to tumors in *Vil-Cre;APC^fl/+^;miR-31^-/-^*mice.

18) Is miR-31 also up-regulated during epithelial regeneration in human intestine (organoids?)? I would like to know its expression status in inflammatory disease patients according to their therapeutic status. Is miR-31 down-regulated when inflammation is well suppressed?

Multiple reports have shown that *miR-31* is upregulated in human inflammatory bowel disease (Beres et al., 2017; Lin et al., 2014; Olaru et al., 2011), as compared to normal intestinal and colonic tissues. We assume that *miR-31* levels go back to normal upon resolution of inflammation, meaning down-regulation of *miR-31*. We agree with the reviewer that it would be very interesting to investigate *miR-31* expression levels according to therapeutic status. However, human inflamed intestinal tissues and their corresponding inflammation-suppressed ones are not available to us using our resource (we attempted, unsuccessfully, to collaborate with local clinical researchers). Thus, at present we cannot confirm this idea directly. We hope that the reviewer can agree with us that this piece of data is outside of the scope of the present study.